# Paper Planes for Teaching Construction Production Systems Based on Lean Tools: Continuous Improvement Cells and 5S

Omar Sánchez [1], María P. Revuelta [1], Adriana Gómez-Cabrera [1] and Luis A. Salazar [2],*

1. Civil Engineering Department, Pontificia Universidad Javeriana, Bogotá 111711, Colombia
2. Departamento de Obras Civiles, Universidad Técnica Federico Santa María, Valparaíso 2390123, Chile
* Correspondence: luis.salazarf@usm.cl

**Abstract:** Teaching production systems and Lean tools is a challenge for educators in the construction area, which is highlighted by the difficulty of emulating in the classroom the scenarios that industry professionals will face. Hence, implementing pedagogical games has a high potential to improve construction education processes. However, gaps are observed in the proposal of pedagogical games applicable to teaching construction systems based on Lean tools. Considering this gap, this paper proposes a game of paper planes to support the teaching of production systems in construction based on the Lean tools Continuous Improvement Cells and 5S. The research method consisted of applying the Design Science Research (DSR) method to develop, evaluate, and improve the game proposal. Thus, the game was assumed as an artifact subject to a development and improvement process to solve an identified problem. The proposed game consists of three main rounds: (1) production system design, (2) Continuous Improvement Cells implementation, and (3) 5S implementation. The main theoretical and practical contribution is the proposal of a pedagogical game to support the teaching of construction production systems based on Lean tools, evaluating students' knowledge by applying surveys before and after the game and assessing performance indicators. The application of the game to a group of undergraduate and graduate students showed a broad positive impact on the assimilation of the principles of construction production systems based on Lean tools.

**Keywords:** Lean manufacturing; Lean construction; construction education; Continuous Improvement Cells; 5S





## 1. Introduction

The construction industry is globally relevant because it employs 7% of the world's working-age population; it also reaches an expenditure in goods and services close to 13% of the world's GDP [1]. However, despite the importance of this sector, this industry has only increased by 1% in the last 20 years [1,2]. This low level of productivity, compared to the rest of the non-agricultural industries, is due to the considerable number of activities that do not add value to the final product [3]. Therefore, the construction industry needs to improve its performance through efficient project management to maximize production value through loss reduction [4].

Organizations continuously implement new management tools and techniques to improve the efficiency of their operations. Among them, Lean principles and tools stand out [5]. On the one hand, in the manufacturing industry, the application of Lean tools such as Just-in-time (JIT) has improved the productivity index [6] by up to 42.08% [7]. On the other hand, in the construction industry, most projects that have implemented Lean principles and tools significantly improve their performance [8]. According to Liker and Meier, Lean Production is a management philosophy based on the Toyota Production System (TPS) [9]. Lean Production promotes collaborative work, worker satisfaction, continuous improvement, and waste elimination. Lean production focuses on value generation by eliminating inefficiencies in the production process (wastes). Some of its main tools,

methods, and approaches are the Last Planner [®] System, Kaizen, 5S, Kanban, Jidoka, Set-Based Design, Linguistic Action Perspective, Choosing by Advantages, Integrated Project Delivery, A3 Report, and Target Value Design, among others [10].

Adopting Lean methodology is important for the Architecture, Engineering, and Construction (AEC) industry [11–16]. Thus, efforts have focused on adapting the curricula of AEC training programs to integrate Lean methodology. However, it has been observed that teaching Lean principles and tools is a challenge for educators, who require pedagogical strategies to complement the theoretical components. As a result, simulations or games are used to teach Lean principles and concepts to engage and empower people, enhancing the learning experience through an applied environment [10,17]. The field of Lean education has benefited from the adoption of games to promote skills development [18]. The games that have been based on Lean tools and principles, some of which have been adopted, are the pig game, parade of trades, role-playing games, paper airplanes, card games, and linguistic exercises, among others [19,20]. In construction management courses, one of the most frequent games is the role-play based on the Last Planner [®] System (LPS) [20,21], in which students assume the role of one of the actors involved in a construction project. This leads the student to reflect on the variables and commitments in the construction project. Hence, the education processes in the AEC industry can benefit from the implementation of games that encourage students to develop skills and reflect on courses' theoretical content [17,21].

Most fields of AEC education tend to rely on laboratories to complement and evaluate the theoretical components provided in the education programs. The structural engineering student designs a beam or column geometry and can rely on laboratory facilities to analyze her decision's impact on the stresses, deflections, or other parameters [22–24]. The geotechnical engineering student can study a soil sample through laboratory tests to complement the theoretical foundations and characterize material properties [25–27]. Similar circumstances occur in other AEC fields, such as hydrology, hydraulics, construction materials, pavements, and others [28–30]. However, using a laboratory so that the construction management student can evaluate the impact of Lean principles and tools is a challenge for educators. This characteristic has led to academic activities having robust theoretical components that to put into practice in the classroom has been a complex task for learners and educators. A difficulty is that the ideal environment to put theoretical knowledge into practice should be an actual construction project or a scenario in which the manager in training could put into practice what he or she has learned and analyze the impact of his or her decisions [31,32].

The adoption of pedagogical strategies that complement the theoretical foundations is crucial for the teaching of Lean principles and tools. The pedagogical games are easy to understand and can be designed for a specific purpose. However, Lean education has been limited to a set of tools and principles, leaving aside the adoption of games for teaching some tools such as the Continuous Improvement Cells and 5S in production systems. Considering this gap, this study aims to propose a game to support the teaching processes of production systems based on the Lean tools Continuous Improvement Cells and 5S. The article is organized as follows. The literature background is provided in Section 2, which contains Lean tool definitions and the literature about games in education. The research method is described in Section 3. Next, the game characteristics are included in Section 4. Then, the results and discussion are divided into two subsections to respond to each research question (see Section 5); the knowledge gaps are analyzed in the discussion section. Finally, conclusions, limitations, and future research are presented in Section 6.

## 2. Literature Background

### 2.1. Continuous Improvement Cells

The diversity of some production procedures, added to volume and demand conditions, has led managers in various industries to divide work into tasks involving specific scopes. The division of labor has caused some organizations to divide the production lines

into groups known as cells, which have been adopted to develop a set of specific tasks in a production process [33]. A cell's organization is based on a manufacturing process focused on producing parts in a line or cell of machines operated by workers linked to the line or cell [34,35]. Therefore, the internal organization of the production cells is characterized by grouping a set of human, equipment, and material resources to contribute to the value generation in the deliverable through developing a set of tasks.

In organizations that adopt cell production systems, the arrangement of the cells is carried out to maintain some elementary principles: each cell is organized to carry out a specific process; each cell has autonomy both in its internal organization and in its decision-making; the cells must be easily adaptable to production conditions; resource inventories in the cells must be sufficient to maintain production continuity, but limited to avoid hindering production processes; communication must be dynamic and efficient between the production cells; the affinity of production times and deliverables is required according to the requirements of subsequent cells that have interdependencies; resources are organized on site considering the continuity and efficiencies of the production process; teams and people are assigned based on the specialty of the cell; and permanent monitoring and control are required to maintain a culture of continuous improvement, among others [34–36]. Figure 1 shows some of the typical production cell organization schemes.

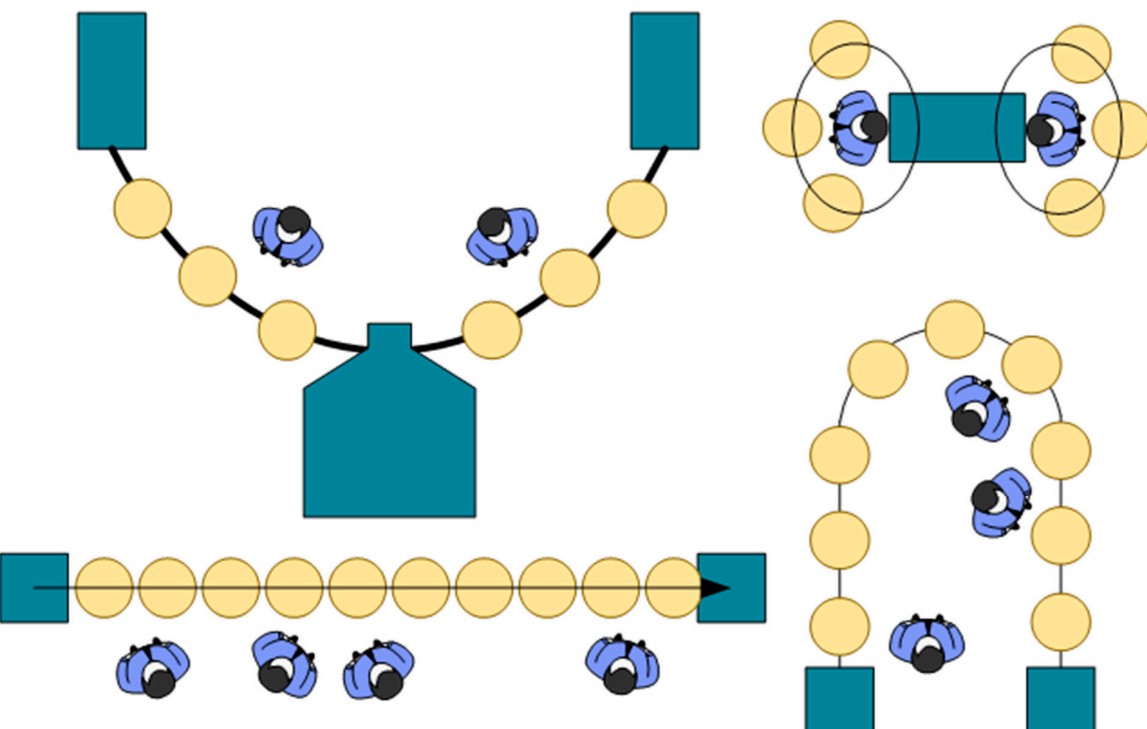

**Figure 1.** Typical production cell organization schemes.

Due to their high degree of specialization and flexibility, the cells are ideal for implementing continuous improvement processes focused on minimizing losses and waste to maximize the value generated in production processes [33,35,36]. Therefore, activities to promote continual improvement can be supported through the adoption of techniques such as problem-solving processes (PDCA cycle), 5S, first run studies, mistake-proofing (poka-yoke) systems, visual management, and others [37]. In the cells, the processes focused on cycles of continuous improvement leads to obtaining more significant benefits that can be used to improve efficiency and quality issues. Thus, the cells have a broad potential to promote improvements in the deliverables through improvements in production processes. In cellular organizations, different benefits have been observed that have contributed to better production processes, such as improvements in productivity, fluid communication,

reduction in production times and inventories, reduction in the unnecessary movement of resources, and specialization of people, among others [33,38–40].

The organization of production processes in cells involves carrying out a set of activities, such as (1) identifying the production processes and their development sequence, (2) selecting the processes with potential for the organization by cells, (3) definition of tasks and deliverables of the processes based on the deliverables, (4) allocation of resources to the selected cells according to the requirements of the related activities, (5) planning the location of the cells according to the restrictions of both space and production processes, (6) organizing people, materials, and equipment according to the sequences of activities identified in each cell, (7) designing the communication and interrelation plan between the cells, (8) monitoring and improving the efficiency of the processes, and (9) promoting a culture of continuous improvement.

The life cycle stages of construction projects involve considerable resources and activities, which means that the organization, management, and improvement of production processes are wasteful tasks. The derived complexity generates adequate scenarios for unwanted events and high impact and recurrence in construction projects, such as delays, cost overruns, disputes, and others [41–47]. Therefore, in the life cycle of construction projects, the adoption of cell production systems has a high potential to contribute to mitigating several factors that generate unwanted phenomena [34]. However, limited adoption of Continuous Improvement Cells in production systems associated with construction projects has been observed, an aspect that highlights the importance of including academic content in the training processes of professionals related to the construction industry.

### 2.2. 5S in Construction

The construction processes are affected by losses and waste generated by supplies disorder and the shortcomings of their storage and location on the work site [48]. Therefore, in the construction industry, there has been a growing adoption of the Lean 5S tool, which is a technique focused on adapting work environments to improve process productivity. Its origin dates to the 1960s when Toyota developed organized and clean workplaces to achieve greater productivity and a better work environment. The 5S tool receives its name according to the initials of the words that represent the stages of the technique in the Japanese language: (1) 'SEIRI' which means sort, (2) 'SEITON' which means set in order, (3) 'SEISO' which means shine, (4) 'SEIKETSUS' which means standardize, and (5) 'SHITSUKE' which means sustain [49–52].

- SEIRI (sort): The initial stage involves a set of actions focused on classifying tools, materials, information, etc., emphasizing the distinction between necessary and unnecessary to add value to production processes. This stage considers the basic needs of the processes, which leads to leaving what adds value and discarding or improving what does not add to or compromises value generation.
- SEITON (set in order): The second stage is based on the results of the first to put in order the items necessary to conduct the activities that generate value and contribute to the deliverables. The organization process is undertaken to ensure that, in any eventuality, any person can resume the activities and find the necessary supplies to carry out their work efficiently. A strategy that is often used consists of adopting as many visual objects as possible to minimize search times during the development of processes [50].
- SEISO (shine): A set of activities that focus on job site cleanliness, which is carried out to keep items and information in the best condition for when they are needed so they are ready for use. The cleaning process begins with a visual inspection of tools, equipment, materials, and other items involved, which aims to identify and remove shortcomings that can compromise the efficiency of the processes and the safety of the workers [53]. The development of this stage usually involves the workers in charge of the procedures to instruct the cleaning activities and their frequency [4,5]. Thus, the

results become inputs for training activities among workers and planning schedules to carry out cleaning activities.

- SEIKETSU (standardize): A process that implies the identification, characterization, and regulation of the normal condition of the work area, which is conducted based on the results of the first three stages. Therefore, this stage leads to the definition of a set of rules to maintain and standardize the identified improvements over time and address abnormal conditions that may occur in the workplace. The process includes the development of a set of guidance documents so that team members can communicate and consult the rules that make it possible to standardize the improvements. Due to its importance, it is recommended that the documentation of the practices be highly descriptive, understandable, and communicative for all team members. Hence, checklists are adopted and are valuable to control the adoption of improvements [51].
- SHITSUKE (sustain): It groups the necessary actions to implement solutions that motivate team members to maintain discipline over time in adopting the improvements obtained from applying the 5S technique. Therefore, this stage entails ensuring that team members have high adaptability and a clear understanding of the correct process structure. This requirement requires managers and facilitators to have leadership skills and knowledge of human resource management [51].

The 5S technique has wide application and the potential to improve various aspects of the life cycle stages of construction projects. At the design stage, adopting the 5S technique can focus on improving information management processes and adapting designers' workstations. This can generate the management of the large volume of information captured, processed, analyzed, and produced by the various disciplines involved in the design of a construction project. Adequate organization of the data allows the design team members to access the appropriate information at the correct time, which helps to mitigate waiting times, shortcomings in the design process, and rework generated from using outdated information.

In the construction stage, the 5S technique can be adopted to improve the management of the use of the workspace and the storage of equipment, tools, and materials [16,54]. Adequate disposal of the supplies of the construction process promotes efficiency in the processes and improves the safety of the workers [6]. These are aspects that are important for construction management if one considers that, on the one hand, the workspace of a construction process is limited, which determines various variables of efficiency and duration of construction activities. On the other hand, construction activities involve elevated levels of risk for workers, who are exposed to multiple scenarios that can lead to accidents, and in the worst cases, to loss of life.

In the operation stage, the 5S technique has applications that may vary depending on the function that the construction project will perform. Organizing and classifying tools and equipment necessary for the project operation will make it possible for operators to carry out maintenance activities more efficiently, which may require personnel who have not previously interacted with the project. Visual aids and organized systems will allow asset managers to understand better the characteristics of the project, which is a crucial aspect in the decision-making processes related to maintenance, adjustments, updates, and other activities necessary in the operation stage.

### 2.3. Games for Education

This section presents different approaches that have been implemented to include games in other educational processes. Subsequently, the implementation of games in construction education is analyzed, and then it is explicitly presented for Lean construction teaching. Finally, includes a review of the main Lean tools implemented in this research: Continuous Improvement Cells and 5S.

### 2.3.1. Games for Construction Education

The literature review searched for articles about construction education using the Web of Science (WoS) and Scopus search engines. The query string was carefully defined, and no year limitation was included, to search for papers including Lean construction and learning construction management in engineering or education in construction. The search terms were (lean construction OR learning construction management OR project based learning) AND (gam * OR active learning OR role-playing game OR serious game) AND (engineering OR education OR collaborative OR construction OR construction engineering) AND (Construction). The search was refined by looking for articles related to the "Architecture," Engineering" and "Construction" areas. Then, a review of the titles and abstracts for each of the papers found in the initial search was developed by screening the selected papers and reporting the impact of implementing games for learning Lean construction. The final selection included 20 articles in the Scopus database and 17 articles in WoS (another six were also in Scopus). A brief map of the literature was developed, analyzing aspects such as the co-occurrence of keywords in titles and abstracts, common terms, the most productive authors, as well as networks of co-authorship. The VOSviewer software was used at this stage (see Figure 2).

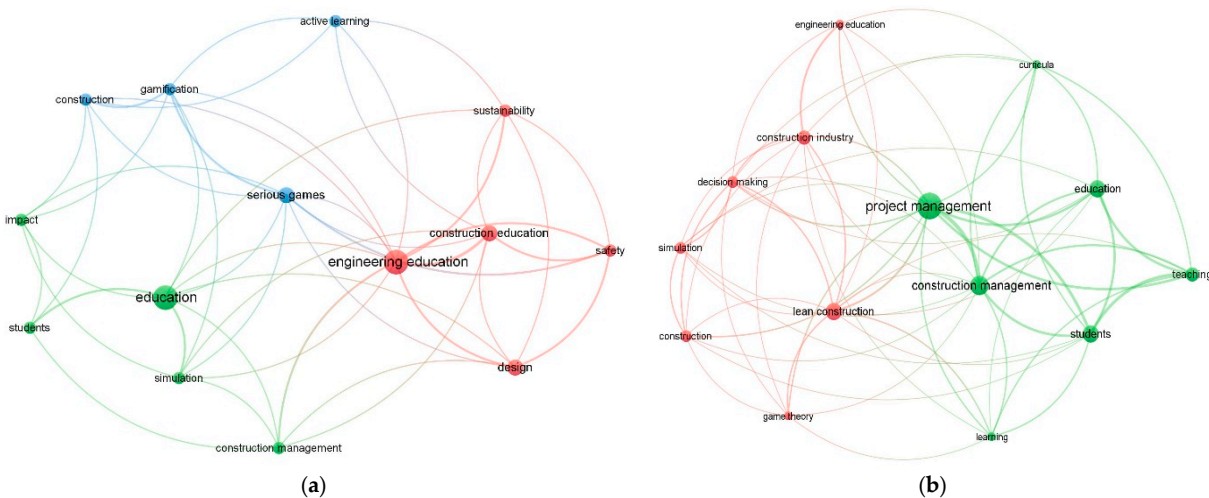

(a)        (b)

**Figure 2.** Most relevant terms in Scopus and WoS. (**a**) Most relevant terms in Scopus; (**b**) Most relevant terms in WoS.

The co-occurrence of all keywords in the VOSviewer software, were filtered for those that appeared at least three times. The 14 most relevant terms that appear at least five times are shown in Figure 2a for the WoS, and in Figure 2b for Scopus. The size of the bubble represents the times the term appeared, and the colors show the clusters according to the co-occurrences, forming three clusters in WoS and two in Scopus. These keywords allowed us to identify that approaches such as simulation, Lean construction, game theory, and active learning have been implemented as engineering education practices in decision-making, sustainability, curricula, and construction management.

Next, considering the citations, the most cited paper included in both databases is LEAPCON: Simulation of Lean Construction of High-Rise Apartment Buildings [55] with 90 citations in Scopus and 40 in WoS. This paper developed a LLean model for the construction management of high-rise apartment buildings with customized apartment designs. A simulated construction process scenario was devised for experimental evaluation of the model in a live management game and the implementation of a discrete event computer simulation of the same process. Results showed that the demonstrative clarity of the Lean model simulation, both live and computerized, makes it a powerful tool for education and research.

The earliest paper was in 2004 [56], in which authors presented a Web-based learning package called the Construction Contracts Information Service, a learning platform as a

way to explore different ways to achieve learning through online simulations; the most recent was in 2022 [57], in which authors developed a learning framework to design a digital simulation game and tested students' perception of its effectiveness in workplace safety and health education obtaining participants high acceptance.

Concerning co-authorship in the WoS database, it was found a cluster of three authors published two papers. Lee et al. [58] developed a framework for the virtual construction simulator game to encourage the broader adoption of simulation games for construction planning and management education; and then the authors discussed the evaluation of this game [59]. In the Scopus database, there was no co-authorship.

### 2.3.2. Lean Games (Simulation)

The literature review identified games related to Lean principles, implemented over time in engineering education, see Table 1. However, among the games identified, some do not explicitly report the Lean principles on which the game is based.

**Table 1.** Games related to Lean Construction principles implemented in Engineering Education.

| Game | Description | L. P. * | References |
|---|---|---|---|
| Lean Airplane simulation | Simulates a production network made of five stations resembling five companies. The game illustrates the benefits of creating flow in a system. It teaches students about teamwork, pulls production, and the impact of supply chain logistics. | No | [60] |
| Silent Squares | It is used to teach students about design management and the importance of sharing incomplete information while applying an integrated design process. It takes the students into the fog of design and shows the challenges designers face requiring and sharing information with other designers. | No | [60,61] |
| Parade of Trades Game | Simulates a construction project with various interdependent contractors. It allows participants to develop a better intuitive understanding of variability and throughput in production systems | No | [60,62,63] |
| Red–Green Simulation | Introduces students to the world of risk management in projects. It teaches them about decision-making and its impact on project stakeholders | No | [60] |
| Helium Stick fame | Teaches students teamwork, cooperation, and good communication to attain a common purpose. | No | [60] |
| DOMEGO | Teaches construction projects to civil engineering students that aim to provide students with active and experiential learning of the key issues of a construction project. In the game, each player embodies a stakeholder of a construction project and must carry out the project while meeting her/his objectives | No | [64] |
| Digital simulation game | Digital simulation game for education. | No | [57,65,66] |
| Serious game | Games developed in an educational environment have additional purposes besides entertainment and fun. | No | [67–69] |
| Virtual Construction Simulator (VCS) | A free and open-source construction management game involves teaching a more holistic decision-making process to plan and manage construction projects. | No | [59] |
| Game theory | An economic game theory model is proposed for understanding the behavior of subcontractors in allocating resources to projects. | No | [70–72] |
| LEBSCO | Simulates aspects of the Last Planner System (LPS) and Lean Production principles. Consists of the assembly of Lego ™ pieces to form a schematic house, and it is played by teams meeting in rounds simulating weeks of work. | Yes | [73] |
| Card Simulation game | Understand the continuous improvement: standardization, pull-system, and collaboration. | Yes | [61] |
| Work Sampling game | Analyze value-added, contributory, and non-contributory activities. | No | [61] |
| Airplane game | Game based on an exercise called "Lean Zone Production Methodologies" create by Visionary Products assembling a Lego ™ airplane. Individuals work in teams and successively introduce Lean principles to their work process during different rounds. | Yes | [74] |
| LEAPCON | A Lean model has been proposed for the construction management of high-rise apartment buildings with customized apartment designs. | Yes | [55] |
| Paper planes | This simulation is based on the process of manufacturing airplanes, to design and improve a production system. | Yes | [75] |
| Villego | A project simulation game was used to teach students and professionals how to use the Last Planner System. It includes constructing a Lego™ structure in different rounds. | Yes | [20] |

* Games that report Lean principles.

*2.4. Paper Airplane Production and Construction Industry*

The production of a paper airplane involves a set of processes with different characteristics and requirements depending on the complexity of the airplane model. Therefore, the manufacturing of a prototype airplane can be divided into activities with defined scopes, which can be organized from a logical sequence focused on the organization of production lines [76]. The characteristics of the production process of a paper airplane, and the low cost required to organize and start up a production line, make the manufacturing process ideal for studying the production processes that may be required in a construction project. For example, in the construction of a reinforced concrete structure, steel elements need to be cast and installed on site. The process involves activities focused on reviewing drawings, selecting materials, measuring, cutting, bending, checking elements, sorting, and installing. This set of activities can be similar to those required in the production of a paper airplane, the difference being that starting up a steel reinforcing elements production line requires specialized personnel and equipment. Those requirements, coupled with the high cost that can be involved in a steel reinforcing element production line, make its adoption in the classroom a challenge for educators. There are other construction production systems that can be assimilated from paper airplane production, such as masonry wall construction, pipe installation, tile construction, ceiling installation, and others. Thus, educators related to the field of production systems in the construction industry can adopt the paper airplane manufacturing process to promote the proposal, analysis, and improvement of construction production systems.

## 3. Research Method

This study focuses on the proposal of a pedagogical game to support teaching processes in construction engineering and management, emphasizing the design of production systems based on Lean tools. Hence, it is necessary to adopt a rigorous research method, which involves systematic activities for designing, developing, evaluating, and adjusting an artifact to provide a solution to an identified need. Considering this requirement, the selected research method was Design Science Research (DSR), which is a rigorous research approach focused on the construction and evaluation of artifacts that are intended to provide useful, innovative, and effective solutions to an existing problem [77,78].

The DSR method makes it possible to contribute both to the theoretical foundation of the artifact and to the solution of the problems for which the artifact is constructed. The main types of artifacts that are usually built by adopting the DSR method are constructs, models, methods, instantiations, and other artifacts where the design leads to a research contribution [79–81]. By focusing on constructing an artifact, the DSR method must involve a set of processes for contextualization, design, and development. In turn, these processes must be complemented with evaluation and feedback processes that are oriented, on the one hand, to improve the characteristics of the artifact to provide an effective solution to the problem that gave rise to the application of the method. On the other hand, it is necessary to evaluate whether the artifact fulfills the function for which it was built [79,82,83]. Therefore, the DSR method involves a continuous improvement process in which a set of cycles are performed to evaluate and adjust the artifact to guarantee its proper functioning and the improvement of its characteristics to provide an efficient solution to the problem.

According to the stages presented by Peffers et al. [84], the research method adopted in this study was divided into five main stages: (1) problem identification, (2) definition of the solution objectives, (3) design and development of the artifact, (4) evaluation and validation of the artifact, and (5) results report. Figure 3 shows the objectives, processes, tools, and analyses that were carried out for each of the research stages.

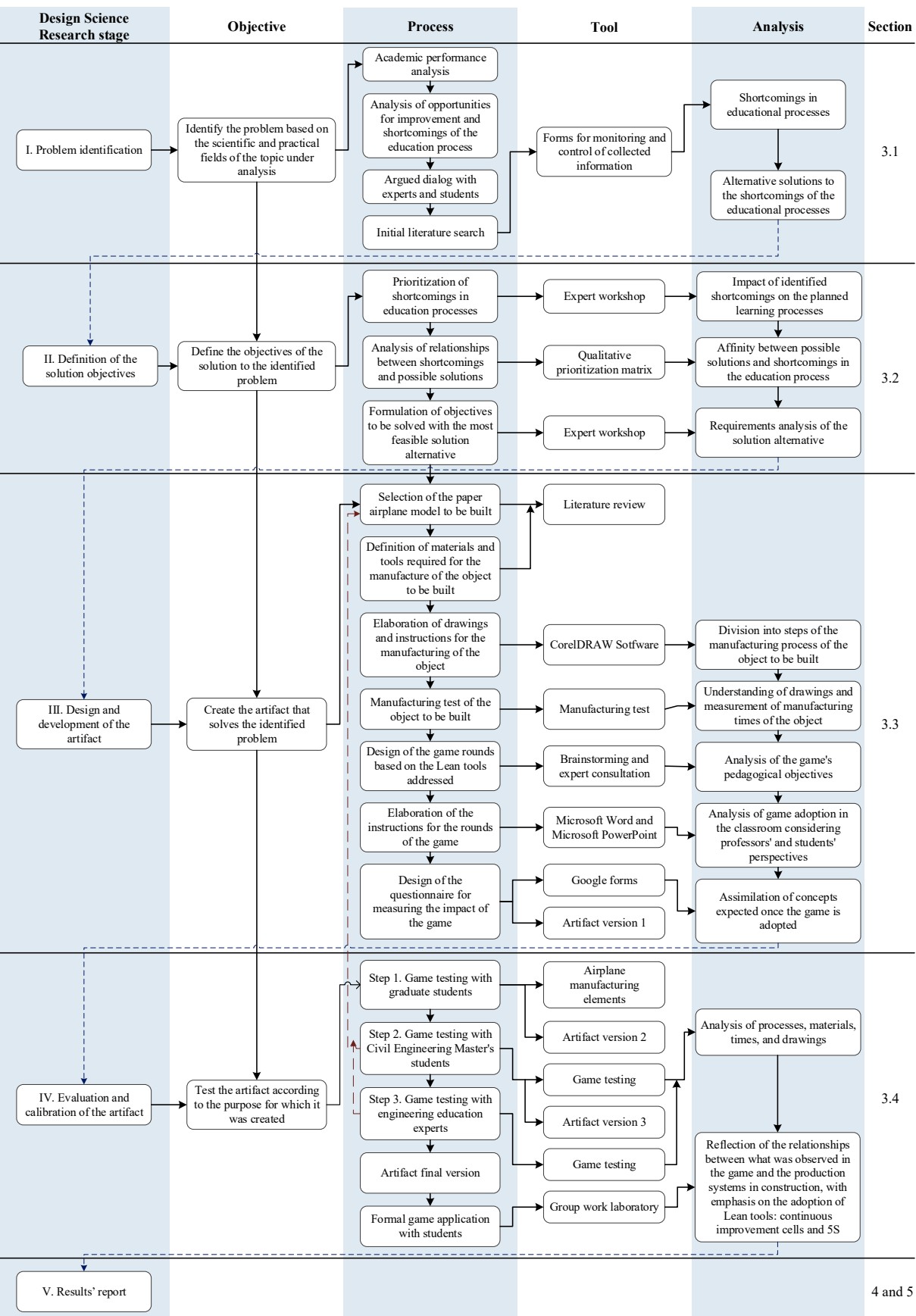

**Figure 3.** Methodological framework.

### 3.1. Problem Identification

The problem identification began with the analysis of the assimilation of production systems fundamentals by Colombian and Chilean undergraduate and graduate students. The analysis was carried out on subjects related to the field of construction management and engineering. The subjects analyzed were construction project management, construction productivity improvement, construction control and scheduling, and design and construction of concrete structures. The analysis was carried out using an expert workshop among four professors with expertise in the field of construction management and engineering (see Table 2). Before the workshop, each professor conducted a consultation exercise with undergraduate and graduate students in his or her charge. The consultation focused on identifying shortcomings in the educational processes related to the teaching of production systems with the implementation of Lean tools and possible improvement actions to address the shortcomings. Based on the findings of the student consultation, the expert workshop was conducted among the four teachers in the field and the following activities were carried out: (1) synthesis of the student's academic performance in the activities related to the topic studied, (2) description of the methodology used to teach the adoption of Lean tools in construction projects, (3) synthesis of successful and unsuccessful experiences in the adoption of teaching strategies, (4) synthesis of suggestions for improvement made by the students, (5) description of shortcomings identified in the teaching processes, and (6 proposal of possible pedagogical strategies to support the process of teaching the adoption of Lean tools in construction projects.

**Table 2.** Professors who participated in the problem identification workshop.

| Id | Profession (Grade) | Research Areas | Years of Teaching Experience |
|----|--------------------|----------------|------------------------------|
| 1 | Civil Engineer, M.Sc., Ph.D. | Construction management and engineering, Lean construction, rural roads, data science in construction | >20 |
| 2 | Civil Engineer, Ph.D. | Construction management and engineering, lean management, Lean construction, engineering education | >10 |
| 3 | Civil Engineer, M.Sc., Ph.D. | Construction management and engineering, Lean construction, building information modeling, integrated project delivery | >10 |
| 4 | Civil Engineer, M.Sc. | Construction management and engineering, Lean construction, building information modeling, engineering education | >5 |

The expert workshop made it possible to identify shortcomings in the educational process and the pedagogical strategies that could contribute to improvement. The main shortcomings or weaknesses identified include the following: (1) lack of applied exercises that address the adoption of Lean tools in construction processes, (2) lack of case studies that show the impact of the adoption of Lean tools in production systems of construction projects, and (3) lack of addressing strategies and methods of implementation of Lean tools in the construction industry. The main pedagogical strategies that could contribute to mitigating the weaknesses or shortcomings were: (1) adoption of pedagogical simulation games in class; (2) adoption of numerical exercises with productivity indexes achieved before and after implementing Lean tools; and (3) assignment of practical group work to be developed during classes, which promotes group reflection on the principles of Lean tools. Based on the findings of the workshop with professors in the field, it was possible to identify the need to implement pedagogical strategies that complement the theoretical components of the courses with practical simulation exercises. In the exercises, students are required to take an active role in measuring indicators that show the impact of the adoption of Lean tools in production systems.

### 3.2. Definition of the Solution Objectives

The development of the second stage began with the analysis of the shortcomings of the education process and pedagogical strategies, which were identified through the expert workshop held in stage 1. The study was carried out through a second workshop with the experts who participated in the first workshop (see Table 2). First, an analysis was made of the possible interactions between shortcomings or weaknesses and pedagogical strategies with potential for improvement. Therefore, each of the experts was asked to analyze how each of the strategies could contribute to mitigating the identified shortcomings. The exercise was carried out with the shortcomings and strategies reported in Section 3.1. Each expert made a reasoned presentation of his or her analysis, emphasizing the requirements for implementing each of the strategies. Second, a general reflection on the strategies with the potential to address the shortcomings was carried out, resulting in a consensus to develop a pedagogical simulation game involving the identified improvement strategies. Third, the experts were asked to select two feasible Lean tools to analyze in conjunction with production systems, considering the 20 Lean tools presented by Tezel et al. [37]: (1) Last Planner System, (2) Visual Management, (3) 5S, (4) Value Stream Mapping, (5) PDCA Cycle, (6) Continuous Improvement Cells, (7) Line of Balance Method, (8) Takt Time Planning, (9) First Run Studies, (10) Work Structuring, (11) Set-up Preparation and Improvement, (12) Supply Chain Integration, (13) Mistake-proofing Systems (Poka-Yoke), (14) In-station quality—Jidoka, (15) Standard Operating Sheets, (16) Just-in-Time (JIT), (17) Pull Production System, (18) Pre-fabrication and Modularization, (19) Cell Production Units, and (20) Information technologies to support Lean construction deployments. Thus, a consensus was reached to implement the tools of Continuous Improvement Cells and 5S.

The workshop ended with the definition of the objectives of the solution to be developed to address the shortcomings or weaknesses of the educational process. Thus, a set of requirements were proposed for the design and development of the pedagogical game: (1) provide the experience of the design and implementation of a production system; (2) confront students with the assignment of roles and responsibilities in a work team; (3) provide scenarios in which it is possible to propose improvement actions to a production system, based on the direct observation and participation of students; (4) make it possible to measure and analyze the impact of the improvement actions proposed to an analyzed production system; (5) provide scenarios to implement and measure the impact on a production system of the continuous improvement cell tools and 5S; and (6) generate the possibility to measure and provide feedback on concepts associated with cycle times, inventories, and Work in Progress.

### 3.3. Design and Development of the Artifact

The design and development of the artifact were based on the requirements defined in stage 2. The design started with a literature review of games for construction education, for which a search was carried out in the Scopus and Web of Science search engines. The literature review identified several games that have been implemented for training on topics related to Lean methodology: Lean Airplane Simulation [60], Parade of Trades Game [60,62,63], Digital Simulation Game [57,65,66], Serious Game [67–69], and Card Simulation Game [61], among others (see Table 1). Despite the variety of games identified in the literature, a lack of games addressing production systems with the implementation of Continuous Improvement Cells and 5S was observed. In addition, it was observed that the Lean Airplane Simulation [60] and Airplane Game [74] could be adapted for teaching production systems that implement the selected Lean tools. The decision to choose the paper airplane as the item to be manufactured was based on the feasibility of making a paper airplane in the classroom, the number of processes required to make a paper airplane, ease of supplying materials and tools necessary, potential motivation among students to test fly the airplane, and ease of quality control of the manufactured object.

A search of the different types of paper airplanes was carried out to define the characteristics of the prototype to be manufactured in the game. Turbines and logos were included

in the defined prototype to increase the number of processes required for its manufacture, which helps students to propose several improvements to the production system. A first outline of the drawings was made to guide the manufacturing process, and the required materials and tools were defined. Next, a manufacturing test was carried out to verify and adjust drawings, materials, and tools.

The artifact's design continued with the design of the game rounds, for which the requirements identified in stage 2 were considered. The artifact design was carried out with the collaboration of the experts who participated in the workshops of stages 1 and 2. It was defined that the game should have three main rounds: (1) production system design, (2) Continuous Improvement Cells implementation, and (3) 5S implementation. The first round was designed for students to propose and implement a production system based on a defined object. The materials and manufacturing tools were limited to encourage the proposal of improvements, and the students were asked to propose the production system based on two-dimensional drawings. It was defined that the materials and manufacturing tools would be delivered without classification or order, which was decided considering the adoption of the 5S tool in a later round. A set of requirements for the airplane to be manufactured were defined, such as the color of the turbines, airline logos, and manufacturing sequence, which was established to increase the complexity and the possibility of improving the production system. The requirements were communicated in a flight schedule that was adapted for each of the rounds of the game (see Appendix C). Thus, five airlines were proposed: (1) Lean Flight (LF), (2) South Airlines (SA), (3) Chile Jet (CH), (4) Colombia Jet (CJ), and (5) Mountain Airlines (MA). The five types of logos were printed with the initial letters of the airlines. Finally, a first version of the description of the instructions was made with the results of the first-round design.

The second round was designed based on the implementation of the Continuous Improvement Cells tool, and the production system obtained in the development of the first round. Therefore, a presentation of the conceptual principles of cell production was included as an activity to be performed by the professor before starting the second round. The plan of the first round was divided so that each participant had detailed instructions on the assigned processes (see Appendices A and B). The division was made considering the segmentation and specialization of the manufacturing processes obtained by adopting cellular production. The quantity and order of materials and tools needed for the manufacturing process were maintained as in the first round.

The third round was designed based on the implementation of the 5S tool and the production system obtained in the previous round. As in the second round, a presentation of the principles of the 5S tool was included before the start of the round. In addition, the possibility was included for students to request additional materials and tools that they considered necessary to promote the efficient adoption of the 5S tool. Thus, students, before starting the round, have the possibility of organizing the workspace with the materials and tools required in the production process. The organization of the space is carried out by adopting the principles of the 5S tool.

The artifact's design ended with the definition of the indicators and the activities to be carried out at the end of each round. The definition of indicators was proposed taking into consideration the usefulness in assimilating concepts related to production systems and comparing what was observed in the three proposed rounds. Table 3 shows the Key Performance Indicators (KPIs) that were selected for measurement in each of the three rounds. The indicators used in this study were based on those proposed by Salazar and Revuelta [76]. A flight test with the airplanes produced according to the defined characteristics was included to support the indicators' measurement and provide a didactic scenario for the students participating in the game. Section 4 presents a detailed description of the instructions for implementing the game.

**Table 3.** Key Performance Indicators.

| Id | KPI | Description |
|---|---|---|
| KPI$_1$ | Production | The total number of airplanes produced correctly. They comply with three rules: (1) the airplane must fly at least 3 m, (2) the airplane must comply with the drawings, and (3) the airplane must comply with the departure itinerary. |
| KPI$_2$ | Cycle Time | Time to complete the first airplane without defects (the airplane must finish the manufacturing process and fly correctly). Time is considered only until the manufacturing process is finished; flight is not considered. |
| KPI$_3$ | Defects | Number of airplanes with failures (not complying with the three rules) |
| KPI$_4$ | Work in Progress (WIP) | Number of incomplete airplanes (manufacturing process started but not completed) |
| KPI$_5$ | Inventory (Stock) | Number of A4 sheets that were not used (the students requested them but did not initiate the manufacture of the airplane) |
| KPI$_6$ | Distance (m) | Sum of flight distances of properly manufactured airplane |
| KPI$_7$ | Performance | (Airplane flown)—(Defective airplane + WIP + Inventory) |

### 3.4. Evaluation and Calibration of the Artifact

The evaluation and calibration of the game were carried out through three main steps: (1) prototype manufacturing test, (2) application of the game to a group of graduate students, and (3) application of the game to a panel of engineering education experts. The first step was the fabrication of the paper airplane prototype from the first version of the drawings. A group of four researchers was assembled and provided with the necessary inputs for the fabrication of the prototype: ruler, scissors, sheets of paper, tape, and logos. A time of three minutes was allotted for the review and analysis of the airplane drawings. Then, the drawings and materials were provided, and the time measurement was started. Thus, it was observed that the time required for a person to produce the paper airplane for the first time varied between 4.3 and 7.5 min. From the airplanes produced, it was observed that three of the four researchers misinterpreted the plans; therefore, the plans were adjusted based on the suggestions made by the researchers. Based on the manufacturing test, the drawings and instructions were adjusted, and it was decided that the time for each round of the game would be nine minutes.

The second step was carried out with the drawings and instructions adjusted according to the observations made in the first step. In the second step, the three rounds proposed for the game were carried out, for which a group of eight students of the Master's Program in Civil Engineering was formed (see Figure 4); the students were selected based on their affinity with the field of construction management and engineering. A period of 18 min was allotted for each round; nine minutes for planning and review of instructions and nine minutes for the production process. According to the drawings and instructions in each round, paper airplane production was zero, four, and ten in the first, second, and third rounds, respectively. At the end of each round, the flight test was performed, which showed that the adequate flight distance to verify production quality was three meters. It was observed that in the third round, the students completed the ten planes of the itinerary in approximately seven minutes. Therefore, the time allotted for each of the rounds was adjusted from nine to seven minutes. At the end of the game, a feedback session was held in which the students made suggestions for improving the game. In addition, an analysis was made of what was observed in the game and its relation to the construction industry. A proposal made by the students was to provide a prototype in the second round, which could be assumed as the BIM model of a construction project. Thus, the drawings supplied in the first round could be considered to be the plans supplied to the contractor at the beginning of the construction process.

The third step was performed with the game adjusted according to what was observed in the second step. A panel was formed of nine experts with extensive experience in engineering education (see Table 4). The four experts who participated in the design and development of the artifact were not included in the group of experts to have a broader view. At the end of the game, there was time for providing feedback. Based on the

expert panel's recommendations, adjustments were made to the construction plans and presentations before starting the second and third rounds, resulting in the final version of the game. Finally, the game was applied in the classes of the subjects analyzed, and the results obtained are presented in Section 5. All students and experts who participated in the study read and accepted the informed consent form (see Appendix F).

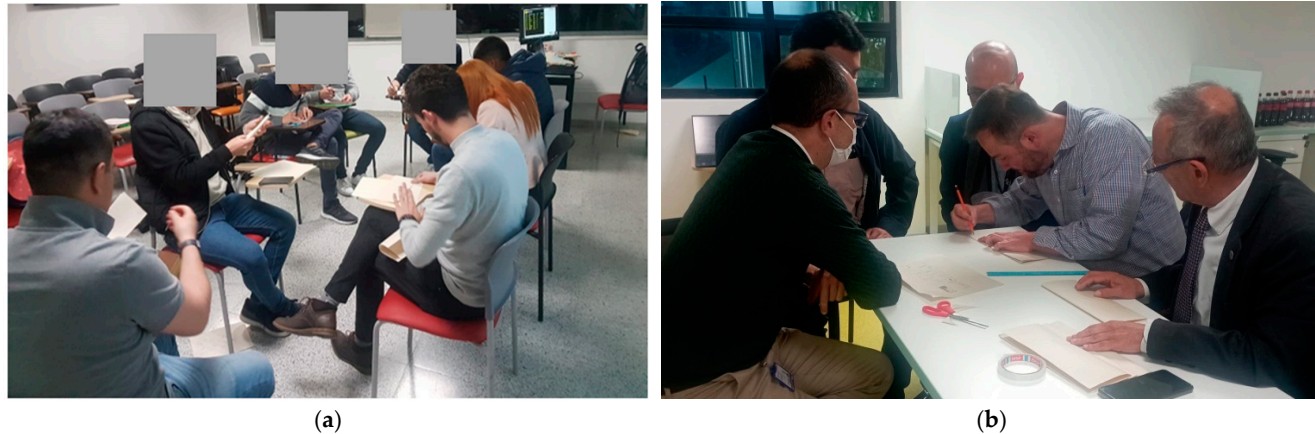

(**a**)                                                                                                        (**b**)

**Figure 4.** Validation and calibration workshops. (**a**) Step 2: Workshop with Civil Engineering Master's students; (**b**) Step 3: Workshop with experts in engineering education.

**Table 4.** Game evaluator professors.

| Id | Profession (Grade) | Research Areas | Years of Teaching Experience |
|----|--------------------|----------------|------------------------------|
| 1 | Civil Engineer, M.Sc., Ph.D. | Materials science and technology, pavements, nanotechnology, road infrastructure | >20 |
| 2 | Civil Engineer, M.Sc., Ph.D. (c) | Structural dynamics, structural materials and systems, structural seismic vulnerability, earth architecture | >20 |
| 3 | Civil Engineer, M.Sc., Ph.D. | Fluvial morpho-dynamics, urban sanitation systems, hydrological and hydraulic modeling, river engineering | >20 |
| 4 | Civil Engineer, M.Sc. | Bridge engineering, seismic vulnerability, ground materials, soil-structure interaction | >20 |
| 5 | Industrial Engineer, M.Sc. | Strategic logistics, social entrepreneurship | >20 |
| 6 | Civil Engineer, M.Sc., Ph.D. | Hydrology, climate change, geographic information systems, remote sensing | >20 |
| 7 | Civil Engineer, Ph.D. | Science and technology of construction materials, construction processes, engineering education | >15 |
| 8 | Civil Engineer, M.Sc., Ph.D. | Mining waste, sanitary landfills, gaseous soils, contaminant control barriers, hydrophobic mineral surfaces, shallow geothermal energy | >10 |
| 9 | Civil Engineer, M.Sc. | Construction management and engineering | >5 |

## 4. Game Characteristics

### 4.1. Game Materials and Tools

The materials needed to implement the game are shown in Table 5. Additionally, the quantities of each material are specified for a course of 21 students, i.e., three groups of players.

**Table 5.** Game materials with quantities for twenty-one students.

| Material | | Quantity | Purpose |
|---|---|---|---|
| Recycled A4 sheets | | 81 | Make paper airplane |
| Scissors | | 6 | Cut out colored sheets to make turbines |
| Tape | | 3 | Glue the turbines |
| A4 Colored sheets | | 9 | Make turbines |
| | LF | 54 | |
| | SA | 36 | |
| Airline logos | CH | 18 | Logo affixed to each side of the airplane |
| | CJ | 36 | |
| | MA | 36 | |
| Rules | | 6 | Measure turbine dimensions and location of turbines and logos |
| Assembled prototype airplane | | 3 | Simulate a 3D model. Support for specific drawings |
| General drawings | | 3 | Drawings with general assembly specifications |
| Specific drawings | | 3 | Drawings with specific assembly specifications |

*4.2. Game Requirements*

For the implementation of the proposed game, some basic requirements are needed:

- The groups should be of seven students, and at least two groups should be formed. Six students will have the roles of airplane assemblers, and one will have the role of controller, in charge of auditing/reviewing the manufacturing process of another group.
- There should be facilitators who are people in charge of recording the KPIs (see Table 3). They can be students or instructors (professors, assistants, and monitors).
- The classroom space should be adequate for the location of the groups of students, materials, and tools according to the designed production cells. In addition, it is recommended to have furniture for the location of materials and implements and to conduct the processes of airplane production. A free space of a minimum area of 70 m$^2$ is recommended for a group of a maximum of twenty-one students.
- Before the application of the game, the instructors must conduct rigorous planning of materials, tools, furniture, space, etc. In this study, the materials and implements shown in Table 5 are proposed, which are estimated for a group of twenty-one students.
- Students answer a questionnaire before and after the activity, the purpose of which is to reflect on the main concepts addressed during the development of the three stages of the game. The final reflection of the activity can be supported by the qualitative and quantitative comparison of the answers obtained in the two questionnaires. In this study, a questionnaire is proposed before and after the application of the game (see Appendices D and E).

*4.3. Game Instructions*

The adoption of the paper airplane game comprises the development of three main rounds: (1) production system design, (2) Continuous Improvement Cells implementation, and (3) 5S implementation. The following recommendations should be considered in the development of the rounds.

- Explain all instructions without giving materials.
- Seven minutes per round.
- Groups of seven participants (one controller and six workers).
- Project or hand out a sheet with the itinerary (see Appendix C).
- The order of manufacture and location on the takeoff line must follow the itinerary, the turbine color specification, and the airline logo.
- All manufacturing steps must be completed for one airplane at a time (according to the drawings), except for the turbines because they are prefabricated. Necessary and sufficient materials (colored sheets and scissors) will be provided for each group to manufacture the turbines of their airplanes.

- Before each round, each group defines the number of planes they plan to make in the 7 min period. The answer determines the number of A4 sheets to give them.
- At the end of each round, the planes are flown, and the instructors record the KPIs (see Table 3).

### 4.3.1. Round 1: Production System Design

The production system design is based on the idealization of a process scheme focused on the production of airplanes, which is the product of the collective participation of the students. For the design of the production system, the following instructions are proposed:

- Assign a space to place: the groups, the messy materials (logos and colored sheets for turbines), and the launch area of the finished airplanes.
- Give general instructions to the groups about the expected product. Hand out a general plan per group, which contains all the steps (see Appendix A) on a single sheet.
- Instruct students to request the number of A4 sheets required to produce the number of planes that they intend to produce in the seven minutes. Additionally, hand out one pair of scissors and one ruler per group.
- Mention that it is allowed to talk but not to plan.
- Explain that the controller must assume the role of supervising the production of an opposing team. The other six members of the group can organize themselves as they wish to produce the airplanes.
- Give the production start signal for all groups.
- Instructors should record the time spent producing each plane. This indicator is recorded per group.
- At the end of the round, record KPIs (see Table 3): check the number of airplanes produced on time, the quality by flying the airplane, and the variability of the product obtained by comparing the product and the specifications supplied.

### 4.3.2. Round 2: Continuous Improvement Cells Implementation

The adoption of the Continuous Improvement Cells is based on the professor's presentation of the theoretical foundations. A set of recommendations is made to divide production into processes with limited scopes. The application of the second round involves the development of the following steps:

- Give a theoretical explanation of cell production.
- Keep the previously defined spaces for the groups, the messy materials (logos and colored sheets for turbines), and the launch area of the finished airplanes.
- Give specific instructions to the groups about the expected product. Hand out five detailed plans per group, which contain the steps divided into five processes (see Appendix B); also, hand out an assembled prototype airplane for reference.
- Explain that each group decides the number of participants in each process. The controller becomes the director (coordinator) of his team and can move his process personnel during the round according to the performance of each one.
- Continue by handing out the requested A4 sheets, one pair of scissors, and one ruler per group.
- Mention that talking is allowed and 7 min are allowed for planning.
- Give the production start signal for all groups.
- Instructors should record the time spent producing each plane. This indicator is recorded per group.
- At the end of the round, record KPIs (see Table 3): check the number of airplanes produced on time, check the quality by flying the airplane, and check the variability of the product obtained by comparing the product and the specifications supplied.

### 4.3.3. Round 3: 5S Implementation

The adoption of 5S is based on the professor's presentation of the theoretical foundations. A set of recommendations is made to organize the workplace to improve productivity. The application of the third round involves the development of the following steps:

- Give a theoretical explanation of Continuous Improvement Cells and 5S.
- Each group can define its workspace and launch area for the finished airplane.
- Continue handing out the requested A4 sheets.
- Hand out the number of scissors and rulers requested by each group.
- Each group organizes the turbine and logos materials in their work area.
- Continue to give specific instructions to the groups on the expected product. Keep the five specific drawings per group and the assembled prototype airplane for reference (see Appendix B).
- The groups continue to decide the number of people in each process. The controller remains the manager (coordinator) of his team and can move his process personnel around during the round based on the performance of each.
- It is still allowed to talk and plan (same seven minutes).
- Give the production start signal for all groups.
- Instructors should record the time spent producing each plane. This indicator is recorded per group.
- At the end of the round, record KPIs (see Table 3): check the number of airplanes produced on time, check the quality by flying the airplane and check the variability of the product obtained by comparing the product and the specifications supplied.

### 4.4. Paper Plane Drawing

The manufacturing of the paper airplanes consists of 11 steps, the first 8 (Figures 5 and 6) correspond to the folds to create the airplane and the placement of the airline logo. The remaining 3 steps (Figure 7) explain the manufacture of the turbines (step 9), the assembly of the turbines (step 10) and the flight of the paper airplane (step 11).

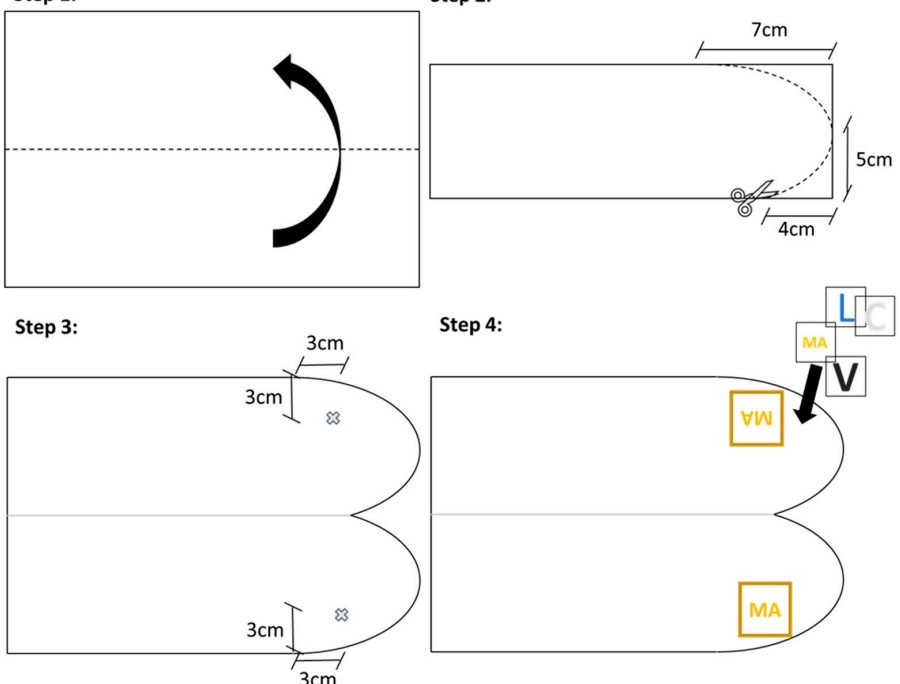

**Figure 5.** *Cont.*

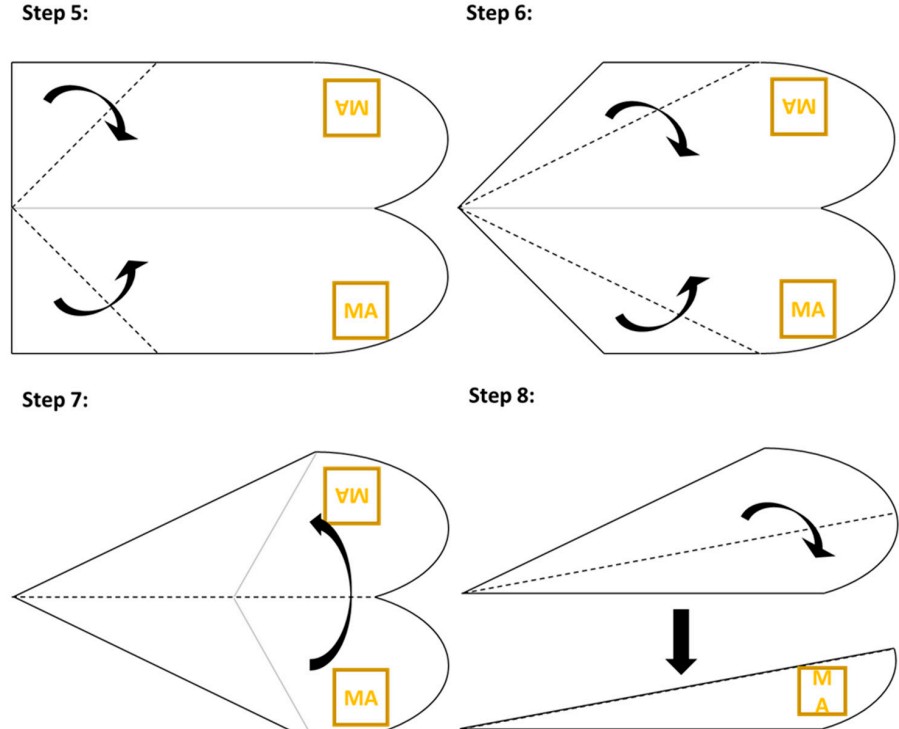

**Figure 5.** Paper plane drawing—Part 1.

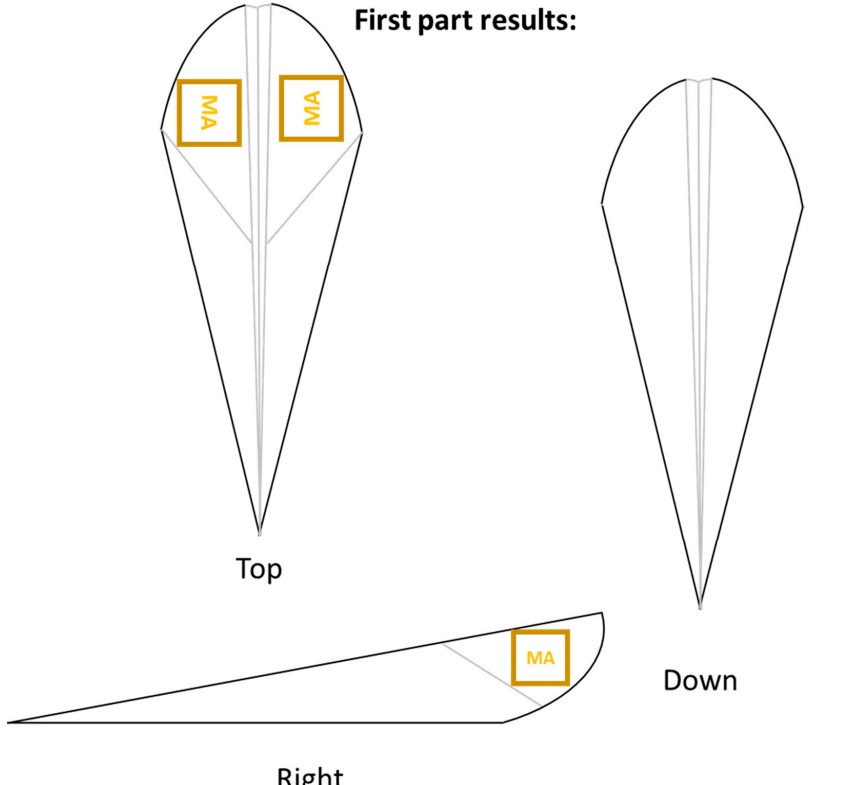

**Figure 6.** Paper plane drawing—Part 2.

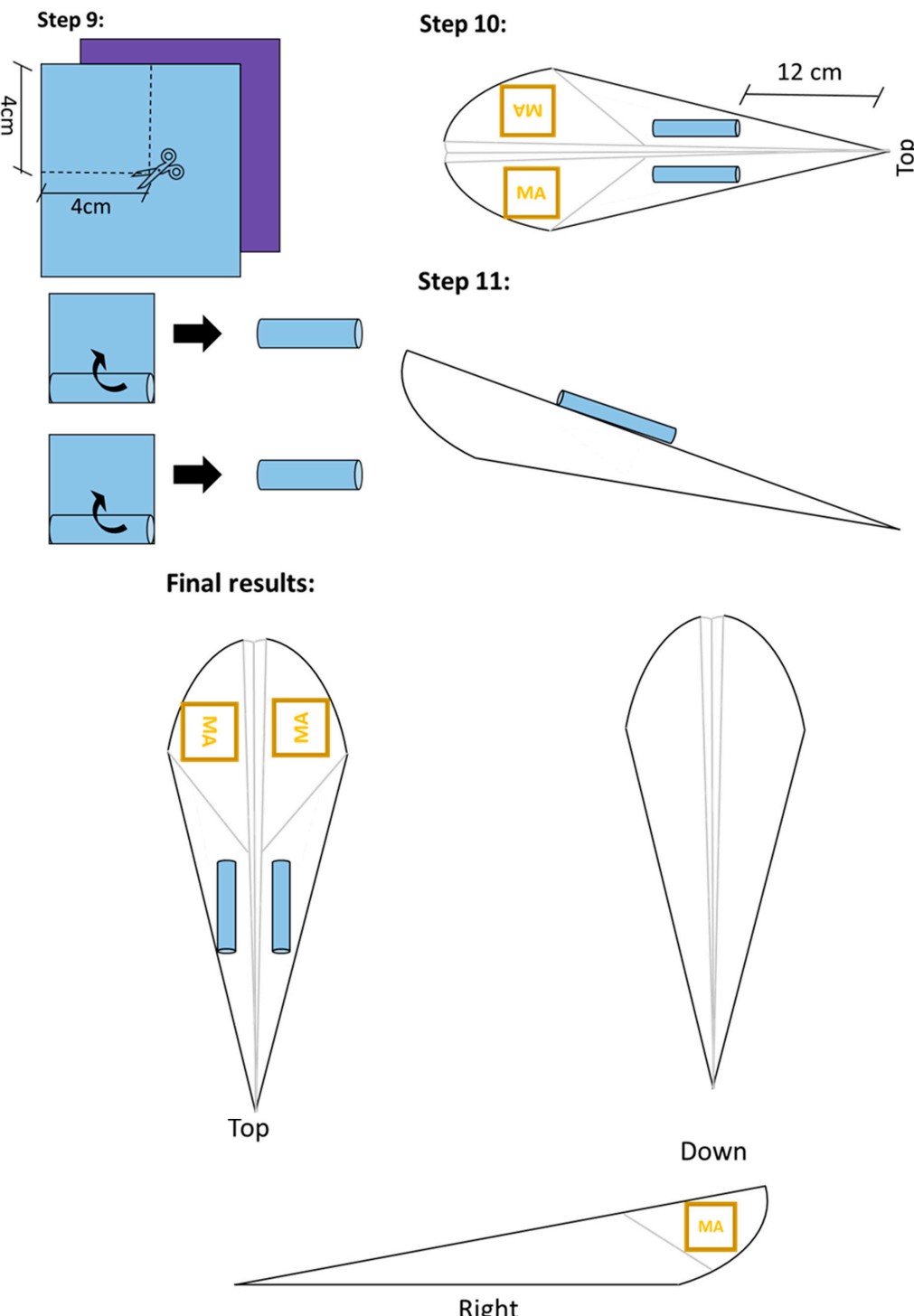

**Figure 7.** Paper plane drawing—Part 3.

## 5. Results and Discussion

The authors applied the game in two South American countries, Chile and Colombia. The game was applied in undergraduate and graduate courses related to the field of construction management and engineering (see Figure 8). Eighty-seven students participated: undergraduate (63) and graduate (24). Each student responded to the pre-game and post-game surveys (see Appendices D and E), and the indicators in Table 3 were measured. This section presents the survey results (pre-game and post-game) and the key performance indicators measured during the game application.

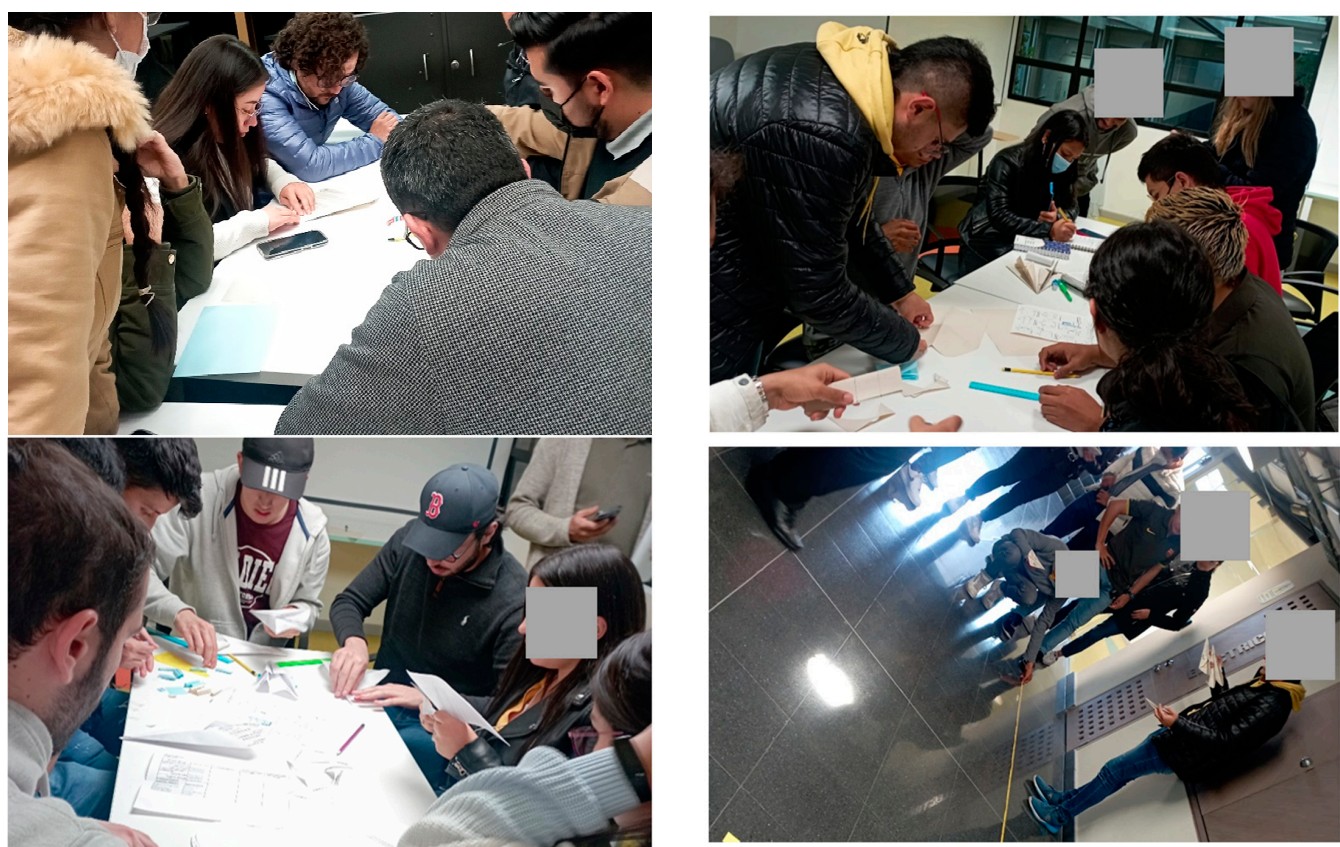

**Figure 8.** Application of the game in undergraduate and graduate courses.

### 5.1. Survey Results

The game and post-game survey results show that the percentage of correct answers increased for all questions in the post-game in both countries (see Figure 9) and both for undergraduate and graduate students (see Figure 10).

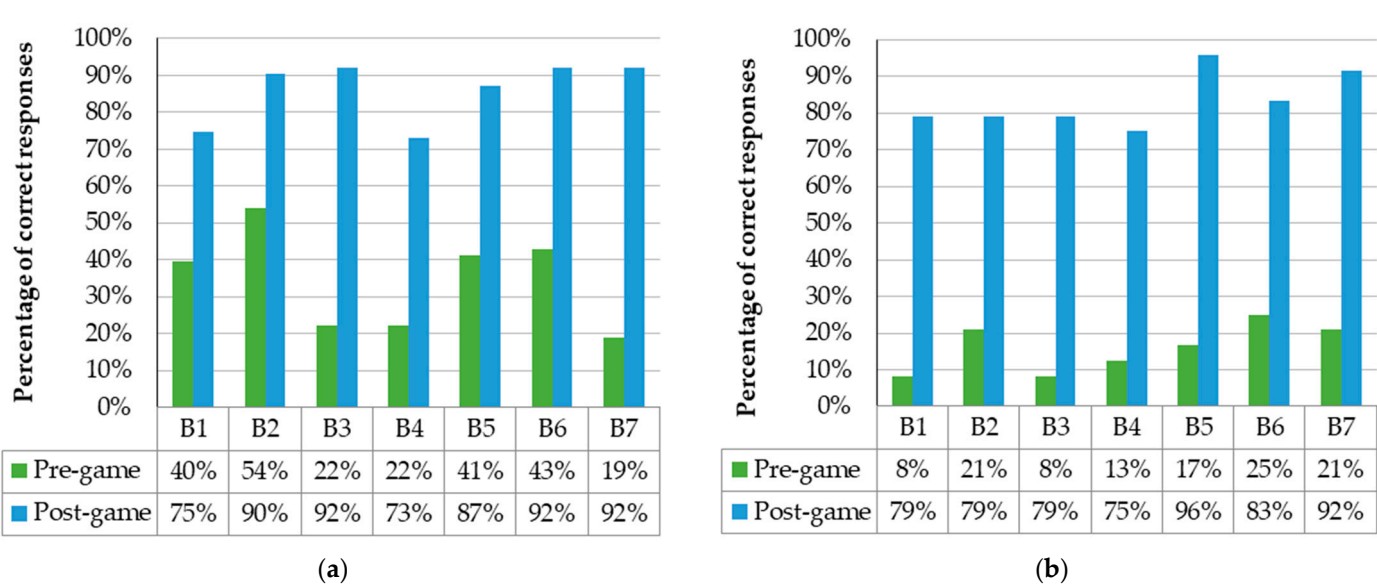

| | B1 | B2 | B3 | B4 | B5 | B6 | B7 |
|---|---|---|---|---|---|---|---|
| Pre-game | 40% | 54% | 22% | 22% | 41% | 43% | 19% |
| Post-game | 75% | 90% | 92% | 73% | 87% | 92% | 92% |

(**a**)

| | B1 | B2 | B3 | B4 | B5 | B6 | B7 |
|---|---|---|---|---|---|---|---|
| Pre-game | 8% | 21% | 8% | 13% | 17% | 25% | 21% |
| Post-game | 79% | 79% | 79% | 75% | 96% | 83% | 92% |

(**b**)

**Figure 9.** Correct pre-game and post-game responses of Colombian and Chilean students. (**a**) Colombian students; (**b**) Chilean students.

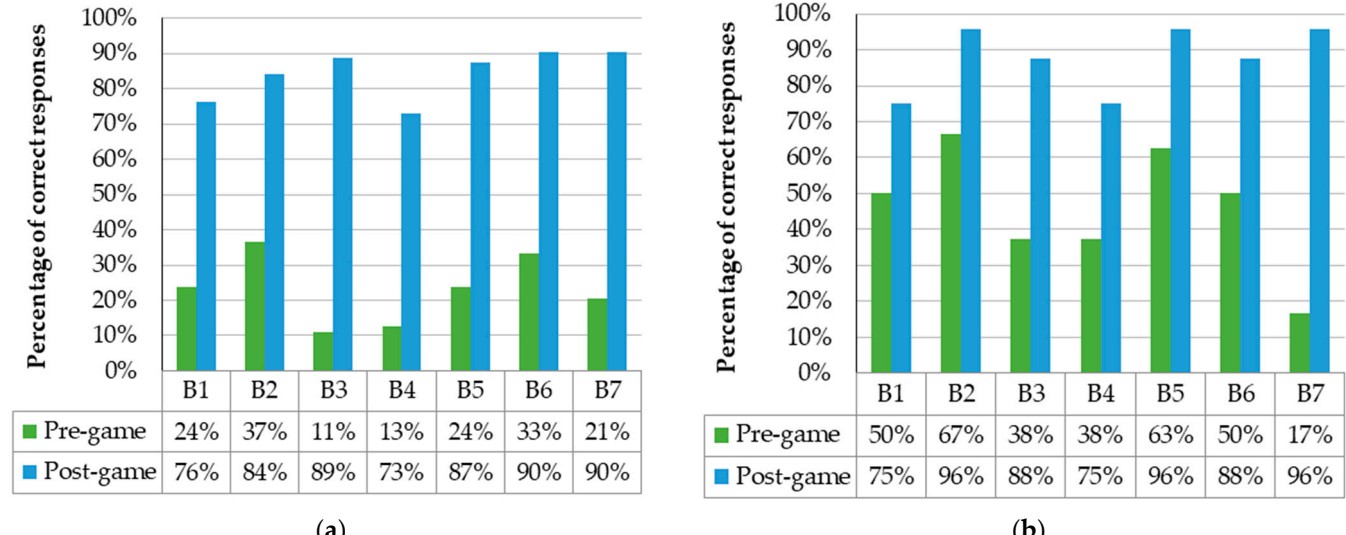

**Figure 10.** Correct pre-game and post-game responses of undergraduate and graduate students. (**a**) Undergraduate students; (**b**) Graduate students.

Figure 9a shows the percentage of correct answers from Colombian students. In the pre-game, the minimum percentage of correct answers was in question B7 (19%) and the maximum in question B2 (54%), while in the post-game, the minimum percentage of correct answers was in question B4 (73%) and the maximum in questions B3, B6, and B7 (92%). Figure 9b shows the percentage of correct answers from the Chilean students. In the pre-game, the minimum percentage of correct answers was in questions B1 and B3 (8%) and the maximum in question B6 (25%), while in the post-game, the minimum percentage of correct answers was in question B4 (75%) and the maximum in question B5 (96%).

Figure 10a shows the percentage of correct answers the undergraduate students. In the pre-game, the minimum percentage of correct answers was in question B3 (11%) and the maximum in question B2 (37%), while in the post-game, the minimum percentage of correct answers was in question B4 (73%) and the maximum in questions B6 and B7 (90%). Likewise, Figure 10b shows the percentage of correct answers from the graduate students. In the pre-game, the minimum percentage of correct answers was in question B7 (17%) and the maximum in question B2 (67%), while in the post-game, the minimum percentage of correct answers was in questions B1 and B4 (75%), and the maximum was in questions B2, B5, and B7 (96%).

Therefore, the researchers agree that in all cases, the game showed an improvement in the level of understanding of the Lean tools when comparing the answers before and after the game. When analyzing all data, the minimum percentage of correct answers was in question B3 in the pre-game and question B4 in the post-game (all figures). On the other hand, as maximum percentages, the correct answers in question B2 in the pre-game and question B7 in the post-game stand out.

The authors wanted to take the next step and analyze the cases with the most significant difference between the results. Figure 11a shows the difference in the percentage of correct answers in the pre-game of undergraduate and graduate students, highlighting an average in undergraduate students of 23% and an average in graduate students of 46%. Therefore, there is a difference between averages of 23%. Figure 11b shows the difference in the percentage of correct answers in the post-game of undergraduate and graduate students, highlighting an average in undergraduate students of 84% and an average in graduate students of 88%. Thus, the difference between averages was 4%.

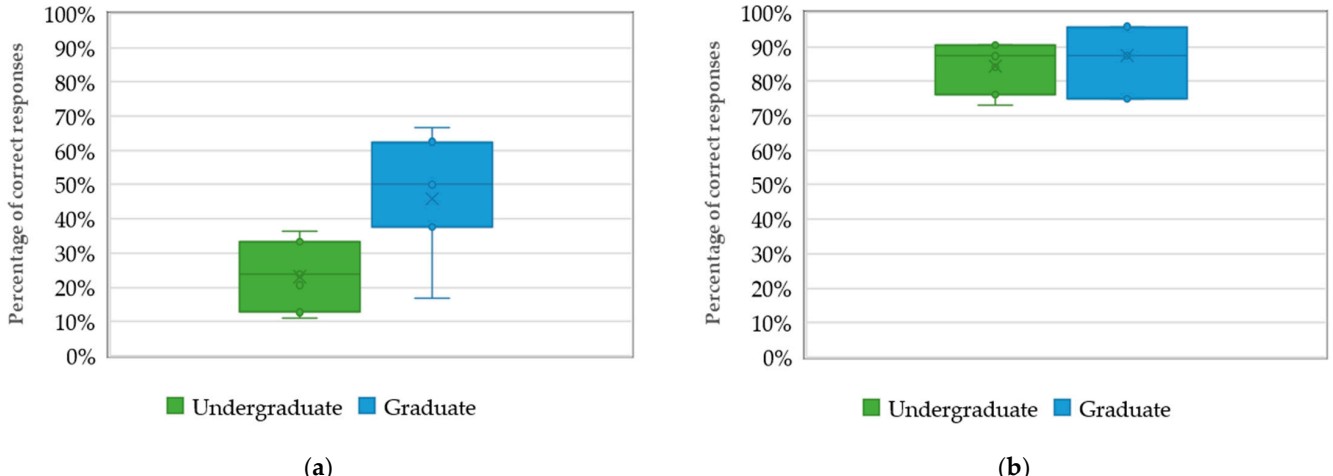

**Figure 11.** Comparison of correct pre-game and post-game responses of undergraduate and graduate students. (**a**) Pre-game correct responses of undergraduate and graduate stu-dents; (**b**) Post-game correct responses of undergraduate and graduate students.

Figure 12a shows the difference in the percentage of correct answers in the pre-game of the Chilean and Colombian students, with an average of 16% for the Chilean students and 34% for the Colombian students. Therefore, there is a difference between averages of 18%. Figure 12b shows the difference in the percentage of correct answers in the post-game of the students from Chile and Colombia, highlighting an average of 83% for the students from Chile and 86% for the students from Colombia. Thus, there is a difference of 3% between averages. Figures 11 and 12 show that the most considerable difference between averages was in the pre-game between undergraduate and graduate students, being 23%. In comparison, the smallest difference was in the post-match between Chilean and Colombian students, with 3%. To statistically analyze the results of the surveys, the researchers developed a bivariate analysis, which consisted of comparing the percentage of correct answers concerning the stage of the survey development. Choosing techniques to implement required the study of the variables' nature. Software RStudio 2022.07.1 was used.

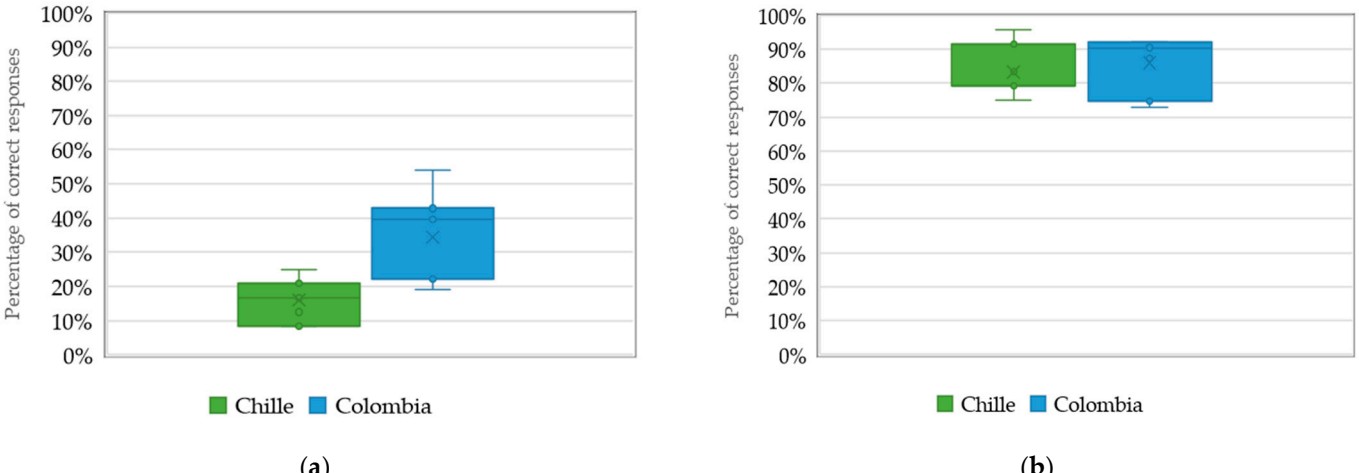

**Figure 12.** Comparison of correct pre-game and post-game responses of Chilean and Colombian students. (**a**) Pre-game correct responses of Chilean and Colombian stu-dents; (**b**) Post-game correct responses of Chilean and Colombian stu-dents.

The first step was to check the normality of the numerical variable with the Shapiro–Wilk tests (more appropriate for small sample sizes <50). The null hypothesis is that a variable is normally distributed [85]. The normality test concluded that data fit a normal distribution

(*p*-value is more than 0.05 (*p*-value = 0.22 for the pre-game stage and *p*-value = 0.10 for the post-game stage); then, a parametric option is needed.

Considering that the dependent variable is numerical (percentage of correct answers), and the independent variables are categorical (stage of survey development) the analysis of variance—ANOVA (parametric) was chosen. The ANOVA test analyzes any difference in the mean values of the groups. The null hypothesis, in this case, is that there is no difference in means [86]. The ANOVA test identifies if groups involved in the categorical variables (pre-game and post-game) present a different behavior concerning the dependent variable analyzed (percentage of correct answers).

The ANOVA test allowed identifying a significant difference between the two groups with a *p*-value of less than 0.05 (*p*-value = $2 \times 10^{-16}$). Graphically, the difference can be seen through a boxplot (Figure 13). For the pre-game stage, the interquartile range (IQR) extends from 0.17, and 0.50, with a median of 0.28. For the postgame stage, the interquartile range extends from 0.78, and 0.93, with a median of 0.87. The better performance is evidenced in the second stage.

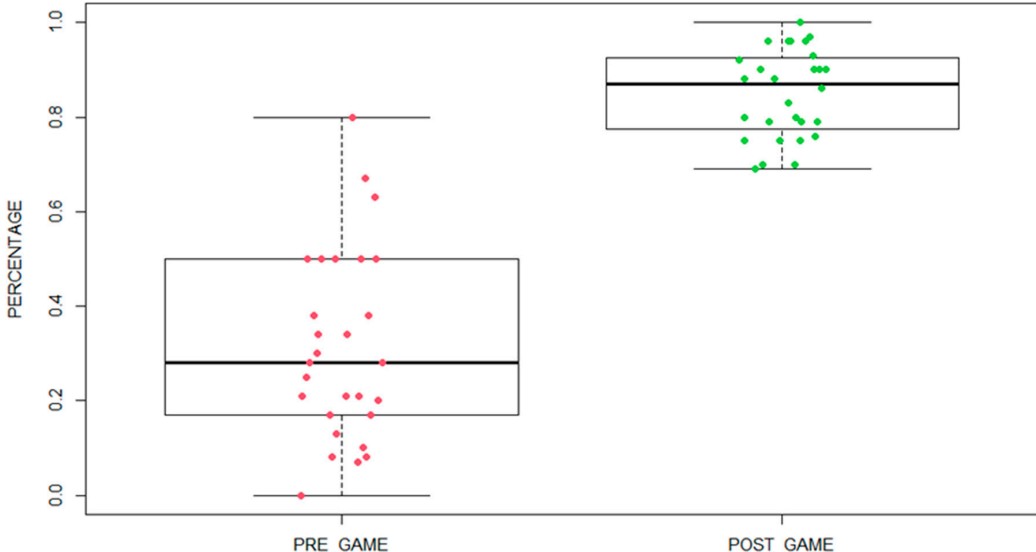

**Figure 13.** Difference between the percentage of correct responses in pre-game and post-game in total students.

For each category, quantile statistics, such as the minimum value, median, maximum, mean, and standard deviation, are shown in Table 6. Better performance is evidenced in the second stage. The minimum of correct answers varies from 0 to 69%, as well as the mean differs from 31% to 85%. Others measurements also show a better performance.

**Table 6.** Pre-game and post-game measurements.

| Measurement | Pre-Game | Post-Game |
| --- | --- | --- |
| Min | 0% | 69% |
| Max | 80% | 100% |
| Mean | 31% | 85% |
| Median | 28% | 87% |
| Standard deviation | 20% | 9% |

Interest groups were analyzed separately by the undergraduate and graduate courses and the countries involved. For all groups, the ANOVA test identified a difference between the two groups (pre-game and post-game). Graphically this information is shown in Figure 14.

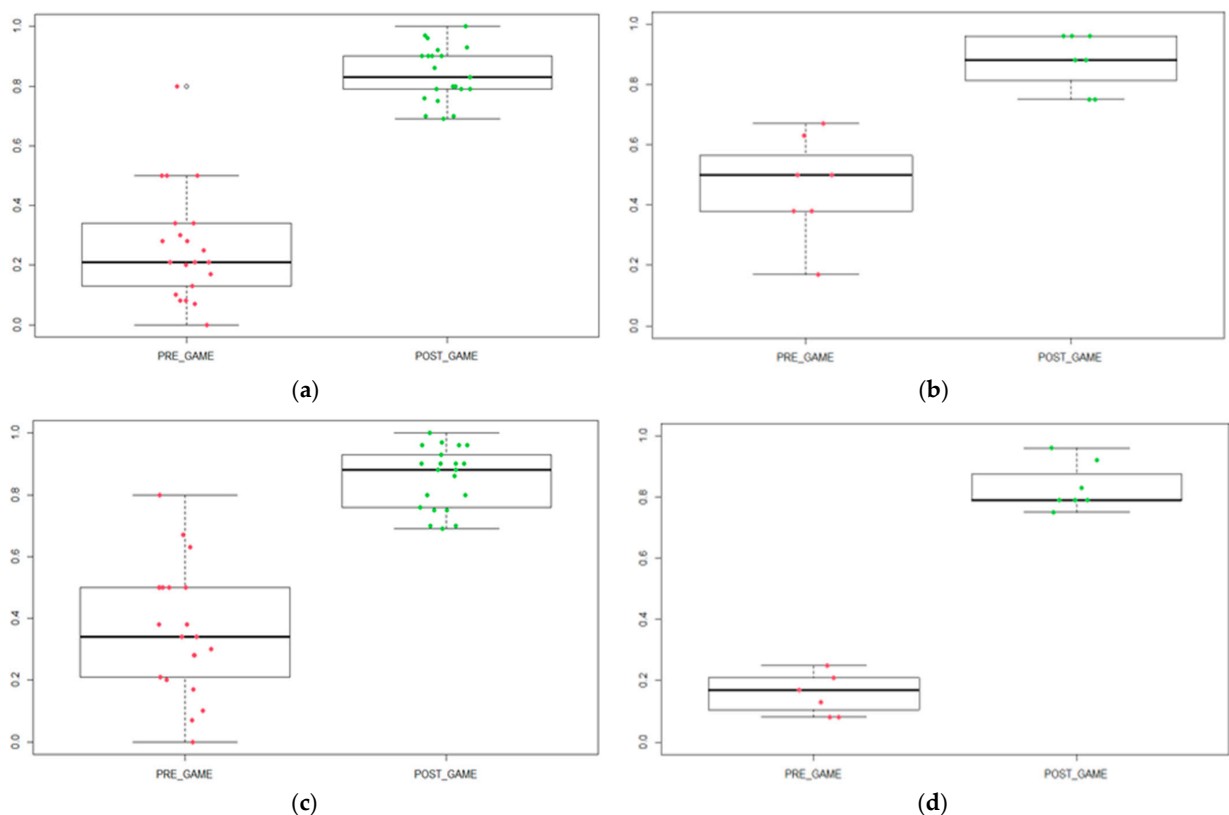

**Figure 14.** Percentage of correct answers in pre-game and post-Game in different scenarios. (**a**) Undergraduate; (**b**) Graduate; (**c**) Colombia; (**d**) Chile.

Finally, the research team analyzed the average percentage of students who responded that they did not know the answer (before and after the game). Figure 15 shows that before the game, the average was 41% (with a minimum of 32% and a maximum of 60%). After the game, the average went to 1% (with a minimum of 0% and a maximum of 2%).

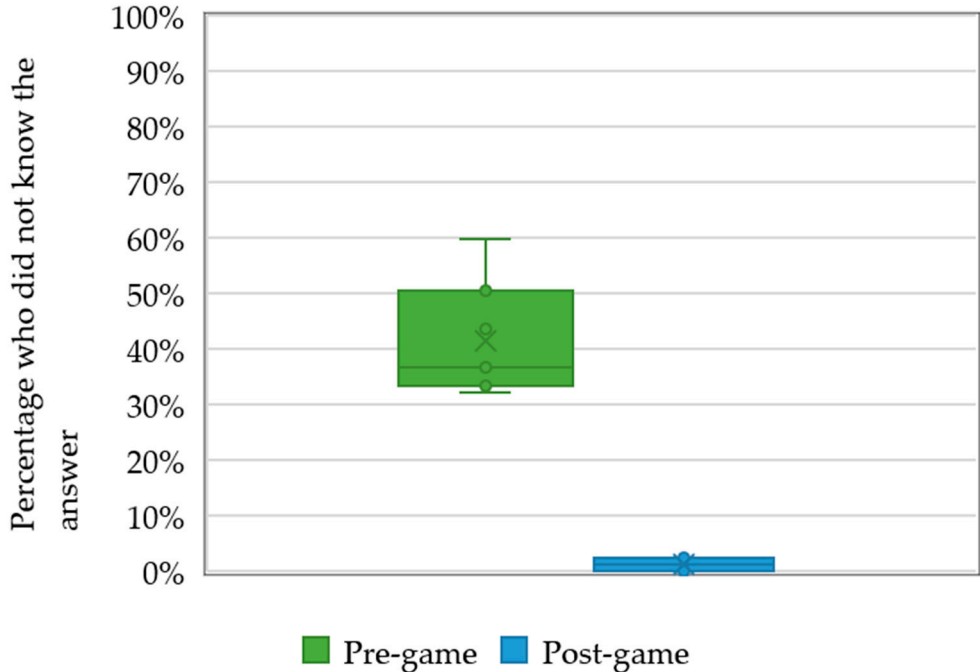

**Figure 15.** Percentage of students who did not know the answer pre- and post-game.

From the above, it could be concluded that the game not only significantly improved the percentage of correct answers but also significantly decreased the declaration of lack of knowledge by the students. Thus, the objective of developing a game that provides an adequate teaching-learning process and supports the teaching of production systems in construction based on the Lean tools: Continuous Improvement Cells and 5S was achieved—improving, in this way, the educational techniques in construction.

### 5.2. Key Performance Indicators Analysis

During the three rounds of the game, the facilitators scored the key performance indicators. At the end of each round, these were presented to the students, analyzing their progress to make the necessary changes to optimize their production. Tables 7 and 8 show the general results, which show a significant improvement in the "performance" indicator for undergraduate students from Colombia and Chile. However, despite an improvement in the result of the performance indicator in the Colombian graduate students, a better result is observed in round 2 than in round 3 (see Tables 7 and 8), because one of the three groups in round 3 produced six airplanes, all of which passed the group's internal quality controls incorrectly, but when they were flown, the facilitators found that all of them had been produced with defects because they did not follow the plans. This unusual factor was analyzed with the students, reaching the conclusion that the group focused on generating many airplanes and downplayed the importance of quality control, believing that they had the experience of the previous rounds; also, the director (coordinator) notified (wrongly) his colleagues that the airplanes arriving at the clearance area complied with the quality because the director misinterpreted the plans and did not check with the prototype (3D model). In conclusion, there was never a warning signal of lack of quality (Jidoka).

**Table 7.** Key performance indicators measured in the application of the game to undergraduate students.

| Undergraduate | | | | | |
|---|---|---|---|---|---|
| Country | Number of Groups | KPI | Round 1 | Round 2 | Round 3 |
| Colombia | 5 | Production (un) | 1 | 14 | 40 |
| | | Cycle Time (minutes) | 7:00 | 2:48 | 2:09 |
| | | Defects (un) | 7 | 11 | 5 |
| | | WIP (un) | 19 | 0 | 0 |
| | | Inventory (un) | 11 | 0 | 0 |
| | | Distance (m) | 3.40 | 80.56 | 217.51 |
| | | Performance | −36 | 3 | 35 |
| Chile | 4 | Production (un) | 1 | 6 | 29 |
| | | Cycle Time (minutes) | 7:00 | 4:56 | 2:33 |
| | | Defects (un) | 1 | 1 | 0 |
| | | WIP (un) | 13 | 14 | 8 |
| | | Inventory (un) | 3 | 0 | 0 |
| | | Distance (m) | 0.00 | 39.00 | 204.40 |
| | | Performance | −16 | −9 | 21 |

**Table 8.** Key performance indicators measured in the application of the game to graduate students.

| Graduate | | | | | |
|---|---|---|---|---|---|
| Country | Number of Groups | KPI | Round 1 | Round 2 | Round 3 |
| Colombia | 3 | Production (un) | 1 | 10 | 14 |
| | | Cycle Time (minutes) | 6:35 | 3:51 | 2:10 |
| | | Defects (un) | 4 | 1 | 7 |
| | | WIP (un) | 9 | 0 | 1 |
| | | Inventory (un) | 2 | 0 | 0 |
| | | Distance (m) | 3.00 | 55.47 | 98.13 |
| | | Performance | −14 | 9 | 6 |

After that, when the authors analyzed the number of airplanes produced and the total flight distance, they observed an improvement in each round in both countries, including undergraduate and graduate students. Finally, analyzing the shortest cycle time by category,

a decrease was also observed in all cases. The rest of the indicators (Defects, WIP, and Inventory) generally decreased (as expected), but present variations that require a study with a larger number of students, including other countries.

With all the above explained, it can be concluded that the proposed game not only significantly improved the percentage of correct answers but also significantly decreased the declaration of lack of knowledge by the students, in addition to showing improvements in the overall performance indicators. In this way, it can be established that the objective of developing a game that achieves an adequate teaching-learning process and supports the teaching of production systems in construction based on the Lean tools: Continuous Improvement Cells and 5S was achieved—improving, in this way, the educational techniques in construction.

## 6. Conclusions

The main contribution of this study is the proposal of a pedagogical game to support the teaching of construction production systems based on the Lean tools: Continuous Improvement Cells and 5S. The game was obtained by applying the Design Science Research method. Therefore, the game is assumed as an artifact that is subject to a process of development and improvement through a rigorous scientific method to provide a solution to an existing problem. The proposed game consists of three main rounds: (1) production system design, (2) Continuous Improvement Cells implementation, and (3) 5S implementation. The first round is carried out to design and implement a production system focused on the manufacture of a defined element. Instructions and materials are supplied, with which the participants have a certain time to design and organize the first scheme of a production system. Next, the participants have a defined time to produce the assigned object. During the production, the moderators measure the indicators that are considered necessary for the pedagogical exercise.

The second round focuses on improving the system proposed in the first round, which is carried out by organizing the production system with the adoption of the Lean Continuous Improvement Cells tool. The moderator makes a presentation about the characteristics of the Lean tool to be implemented. The participants will have a defined time to organize the production system according to the principles of cellular production. Next, the participants will have a time equal to that of the first round to produce the assigned object. The moderators measure the indicators defined in the first round during the production process.

The third round is carried out from the production system obtained in the second round, in which the Lean 5S tool is implemented. As in the second round, the moderator makes a presentation about the characteristics of the Lean tool. Then, the participants will have a defined time to modify the production system from the 5S tool and what was observed in the production process of the second round. As in the previous rounds, the moderators measure the indicators. Finally, with the results of the three rounds, the moderator directs the participants to reflect on what was observed in the development of the game. The pre-game and post-game survey results show that the percentage of correct answers increased for all questions in both countries and both undergraduate and graduate students. Therefore, the researchers agreed that, in all the exposed cases, the game showed an improvement in the level of understanding of Lean tools when comparing the answers before and after the game. It could be concluded that the proposed game not only significantly improved the percentage of correct answers but also significantly decreased the students' statements of lack of knowledge. Subsequently, when the authors analyzed the number of aircraft produced and the total flight distance, they observed an improvement in each round in both countries, including undergraduate and graduate students. Finally, when analyzing the shortest cycle time by category, a decrease was also observed in all cases. In this study, a paper airplane was used as the object to be produced; however, educators can adapt the proposed game to other objects to be produced depending on the objectives of the course in which the game is to be implemented.

Limitations of this study are: (1) the lack of integrating Lean tools other than Continuous Improvement Cells and 5S into the proposed game; (2) the application of the game to a limited group of Colombian and Chilean students; (3) the adoption of only the paper airplane as an object to be built; (4) the lack of tests focused on the application of the game with different times, number of participants, rules, tools, and materials. Therefore, future work could be focused on: (1) the proposal of pedagogical games to support the teaching of Lean tools different from those studied; (2) the application of the proposed game to groups of students of different nationalities and levels of training, with which it would be possible to analyze and compare the reported indicators; (3) the application of the game with the modification of the object to be built, which can focus on an object that involves different materials and tools; (4) the proposal of modifications to the game presented from the variation of the game characteristics, such as times, number of participants, rules, and others.

**Author Contributions:** Conceptualization, O.S., M.P.R., A.G.-C. and L.A.S.; methodology, O.S., M.P.R., A.G.-C. and L.A.S.; results analysis, O.S., M.P.R., A.G.-C. and L.A.S.; writing—original draft preparation, O.S., M.P.R., A.G.-C. and L.A.S.; writing—review and editing, O.S., M.P.R., A.G.-C. and L.A.S.; visualization, O.S., M.P.R., A.G.-C. and L.A.S.; supervision, O.S., M.P.R., A.G.-C. and L.A.S. All authors have read and agreed to the published version of the manuscript.

**Funding:** This research received no external funding.

**Institutional Review Board Statement:** Ethical review and approval were waived for this study due to the study is part of a pedagogical strategy adopted by professors to improve the teaching of production systems in construction. The activities performed are aligned with the Declaration of Helsinki. Each student and professor participating in the pedagogical game agreed with the informed consent (see Appendix F).

**Informed Consent Statement:** Informed consent was obtained from all subjects involved in the study (see Appendix F).

**Data Availability Statement:** Not applicable.

**Acknowledgments:** Omar Sánchez thanks Colciencias for the sponsorship and support through the "Convocatoria Doctorados Nacionales—2015" program. Colciencias is the Administrative Department of Science, Technology, and Innovation, a Colombian government agency that supports fundamental and applied research in Colombia.

**Conflicts of Interest:** The authors declare no conflict of interest.

## Appendix A. General Drawings

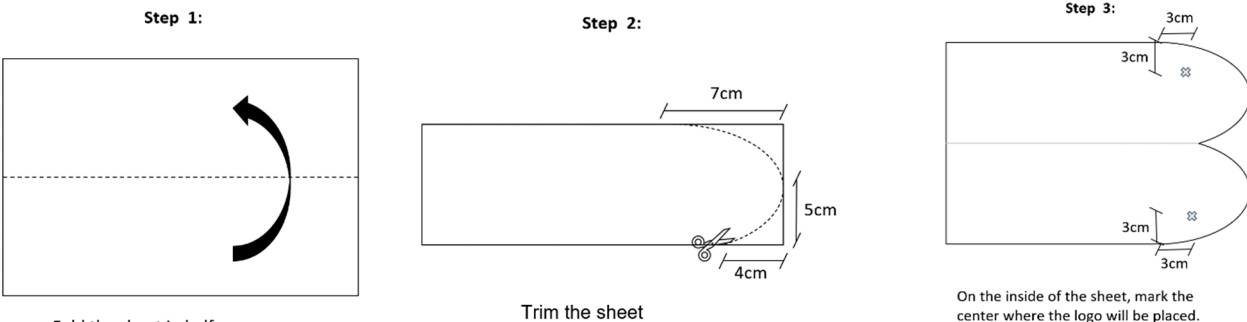

**Step 1:**

Fold the sheet in half

**Step 2:**

7cm

5cm

4cm

Trim the sheet

**Step 3:**

3cm

3cm

3cm

3cm

On the inside of the sheet, mark the center where the logo will be placed.

**Figure A1.** *Cont*.

**Step 4:**

Place the airline logo in the center of the mark. Keep in mind the itinerary to select the airline.

**Step 5:**

Fold the corners of the sheet inward.

**Step 6:**

Fold inward according to the plan.

**Step 7:**

Fold inward.

**Step 8:**

Fold outward

**Step 9:**

4cm

4cm

Make 2 turbines for the airplane (same color according to itinerary) by cutting out 4x4cm squares and making cylinders.

**Step 10:**

12 cm

Top

Tape the turbines to the top of the plane.

**Step 11:**

Locate the aircraft on the takeoff line following the itinerary.

**Final results:**

Top

Down

Right

**Figure A1.** Paper plane drawings.

## Appendix B. Specific Drawings

### Process 1

**Step 1:**

Fold the sheet in half

**Step 2:**

7cm

5cm

4cm

Trim the sheet

**Step 3:**

3cm

3cm

3cm

3cm

On the inside of the sheet, mark the center where the logo will be placed.

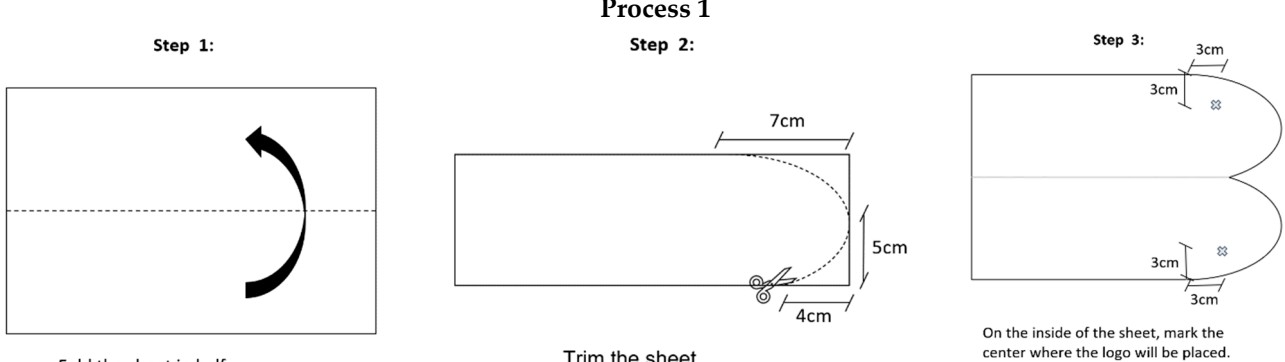

**Figure A2.** *Cont.*

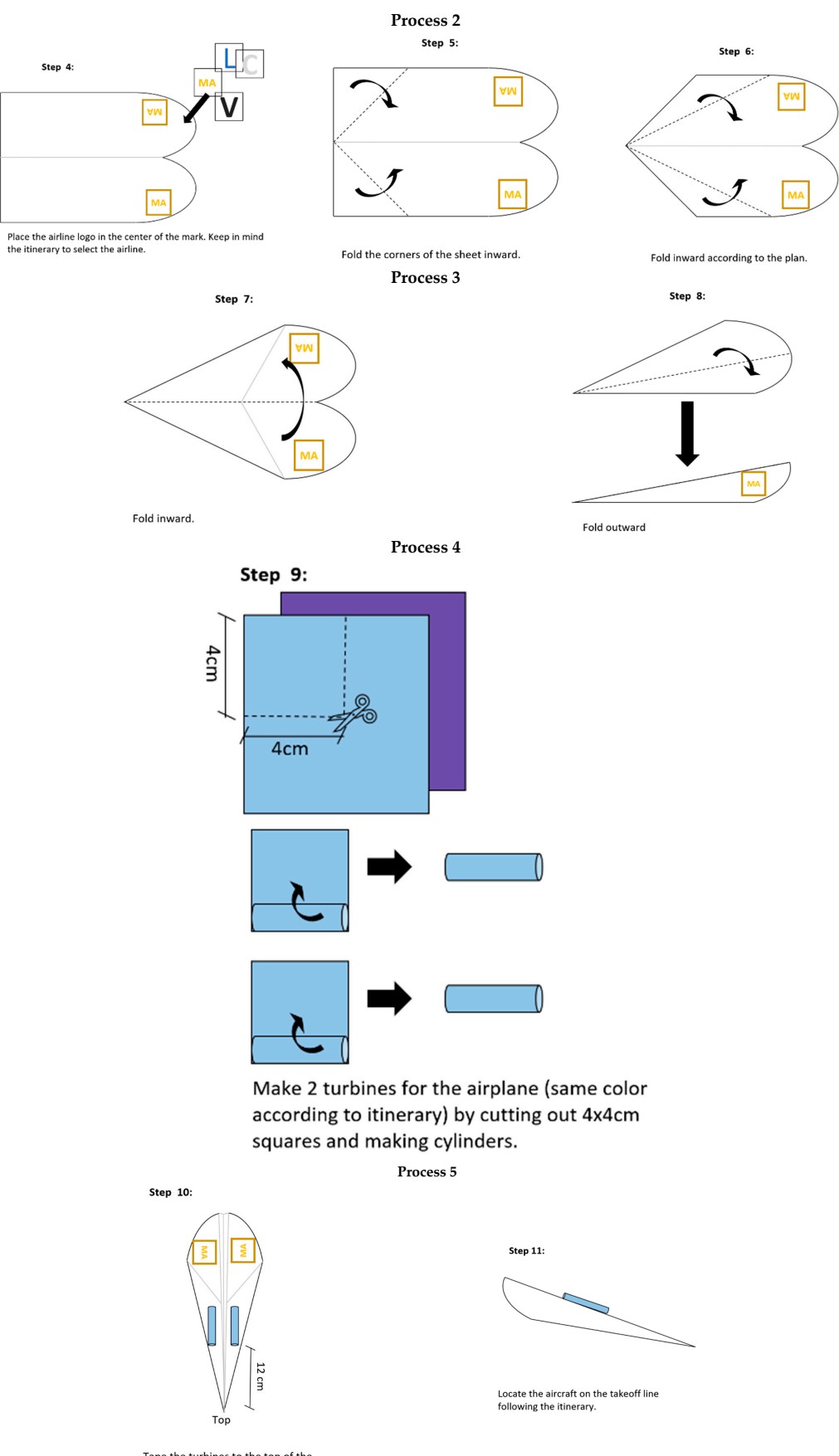

**Figure A2.** Paper plane specific drawings.

**Appendix C. Itinerary**

Round 1:

| TIME | DESTINATION | FLIGHT | GATE | REMARKS | COLORS |
|------|-------------|--------|------|---------|--------|
| 12:33 | PARIS | LF 147 | 31 | BOARDING | Blue |
| 12:54 | TOKYO | SA 547 | 27 | CANCELLED | Yellow |
| 13:04 | BERLIN | CH 764 | 35 | DELAYED | Blue |
| 13:22 | NEW YORK | CJ 982 | 14 | | Yellow |
| 13:47 | MADRID | MA 142 | 12 | | Blue |
| 14:03 | ROME | SA 395 | 9 | | Yellow |
| 14:23 | SYDNEY | LF 734 | 33 | | Blue |
| 14:55 | BARCELONA | LF 762 | 24 | | Yellow |
| 15:01 | TORONTO | CJ 336 | 26 | | Blue |
| 15:15 | MOSCOW | MA 825 | 13 | | Yellow |

Round 2:

| TIME | DESTINATION | FLIGHT | GATE | REMARKS | COLORS |
|------|-------------|--------|------|---------|--------|
| 12:33 | NEW YORK | MA 142 | 31 | BOARDING | Blue |
| 12:54 | MADRID | SA 395 | 27 | CANCELLED | Yellow |
| 13:04 | BARCELONA | LF 734 | 35 | DELAYED | Blue |
| 13:22 | TOKYO | MA 825 | 14 | | Yellow |
| 13:47 | ROME | LF 147 | 12 | | Blue |
| 14:03 | SYDNEY | SA 547 | 9 | | Yellow |
| 14:23 | SYDNEY | CH 764 | 33 | | Blue |
| 14:55 | PARIS | CJ 982 | 24 | | Yellow |
| 15:01 | TORONTO | LF 762 | 26 | | Blue |
| 15:15 | MOSCOW | CJ 336 | 13 | | Yellow |

Round 3:

| TIME | DESTINATION | FLIGHT | GATE | REMARKS | COLORS |
|------|-------------|--------|------|---------|--------|
| 12:33 | SYDNEY | CH 764 | 31 | BOARDING | Blue |
| 12:54 | PARIS | CJ 982 | 27 | CANCELLED | Yellow |
| 13:04 | TORONTO | MA 825 | 35 | DELAYED | Blue |
| 13:22 | MOSCOW | CJ 336 | 14 | | Yellow |
| 13:47 | MADRID | LF 147 | 12 | | Blue |
| 14:03 | BARCELONA | SA 547 | 9 | | Yellow |
| 14:23 | ROME | MA 142 | 33 | | Blue |
| 14:55 | SYDNEY | SA 395 | 24 | | Yellow |
| 15:01 | NEW YORK | LF 734 | 26 | | Blue |
| 15:15 | TOKYO | LF 762 | 13 | | Yellow |

**Figure A3.** Itineraries.

**Appendix D. Pre-Game Survey**

Evaluation of the teaching-learning process of the paper plane game.

A1.    Enter the code to answer the pre-game form. The format should be the first letters of the first and last name plus the day and month of birth (FLDDMM). Example: Luis Salazar 08/23 = code LS2308.

Answer: ______

A2.    Select your educational institution.

a.    Pontifical Javeriana University—Colombia.
b.    Federico Santa María Technical University—Chile.

A3.    Select your level of education.

a.    Undergraduate.
b.    Graduate.

A4.    Were you a game facilitator?

a.    Yes.
b.    No.

B1.    Which of the following alternatives is the best definition of "Continuous Improvement Cells"?

a.    Gradually improve the performance of a project through intensive planning.
b.    Divide the work into groups with tasks involving specific scopes within a production process.
c.    Technique focused on adapting work environments to make them organized and clean, thus improving the productivity of the processes.
d.    Exponentially improve the performance of a project through process mapping.
e.    I don't know the answer.

B2.    Which of the following alternatives corresponds to the basic principles of Continuous Improvement Cells?

a.    Each cell does not have autonomy as regards its internal organization and decision-making.
b.    Teams and people are assigned based on the cell specialty.
c.    Continuous monitoring and control are not required to maintain a culture of continuous improvement.
d.    Resources are allocated equitably without distinction between processes and ranks of participants.
e.    I don't know the answer.

B3.    Which of the following alternatives corresponds to the best definition of "5S"?

a.    Technique focused on adapting work environments to make them organized and clean, thus improving productivity.
b.    A method that allows finding the root cause of productivity problems.
c.    Process of continuous improvement based on concrete, simple and inexpensive actions involving all the workers of an organization.
d.    Five ways to select industrial processes.
e.    I don't know the answer.

B4.    Which of the following alternatives corresponds to the correct order of application of "5S"?

a.    SEISO (shine)—SEIRI (sort)—SEITON (set in order)—SEIKETSU (standardize)—SHITSUKE (sustain).
b.    SEISO (shine)—SEITON (set in order)—SEIRI (sort)—SEIKETSU (standardize)—SHITSUKE (sustain).

    c.    SEIRI (sort)—SEITON (set in order)—SEISO (shine)—SEIKETSU (standardize)—SHITSUKE (sustain).

    d.    SEIRI (sort)—SEITON (set in order)—SEISO (shine)—SHITSUKE (sustain)—SEIKETSU (standardize).

    e.    I don't know the answer.

B5.    Which of the following alternatives corresponds to the definition of "cycle time"? (Please consider the airplane manufacturing process).

    a.    Time required to produce the total number of airplanes without defects.

    b.    Time required to produce one cycle (10 airplanes) without defects.

    c.    Time required to produce the first defect-free airplane.

    d.    None of the answers is correct.

    e.    I don't know the answer.

B6.    Which of the following alternatives corresponds to the definition of "WIP (Work in Progress)"? (Please consider the airplane manufacturing process).

    a.    A number of incomplete airplanes (progress was made, but not completed).

    b.    A number of airplanes were completed but with defects (these did not fly).

    c.    A number of unused sheets (requested but not used).

    d.    None of the answers is correct.

    e.    I don't know the answer.

B7.    Which of the following alternatives corresponds to the definition of "Inventory"? (Please consider the airplane manufacturing process).

    a.    A number of incomplete airplanes (progress was made, but not completed).

    b.    A number of airplanes were completed but with defects (these did not fly).

    c.    A number of unused sheets (requested but not used).

    d.    None of the answers is correct.

    e.    I don't know the answer.

**Appendix E. Post-Game Survey**

Evaluation of the teaching-learning process of the paper plane game.

A1.    Enter the code to answer the pre-game form. The format should be the first letters of the first and last name plus the day and month of birth (FLDDMM). Example: Luis Salazar 08/23 = code LS2308.

Answer: ______

A2.    Select your educational institution.

    a.    Pontifical Javeriana University—Colombia.

    b.    Federico Santa María Technical University—Chile.

A3.    Select your level of education.

    a.    Undergraduate.

    b.    Graduate.

A4.    Were you a game facilitator?

    a.    Yes.

    b.    No.

B1.    Which of the following alternatives is the best definition of "Continuous Improvement Cells"?

    a.    Gradually improve the performance of a project through intensive planning.

    b.    Divide the work into groups with tasks involving specific scopes within a production process.

    c.    Technique focused on adapting work environments to make them organized and clean, thus improving the productivity of the processes.

      d.      Exponentially improve the performance of a project through process mapping.

      e.      I don't know the answer.

B2.     Which of the following alternatives corresponds to the basic principles of Continuous Improvement Cells?

      a.      Each cell does not have autonomy as regards its internal organization and decision-making.

      b.      Teams and people are assigned based on the cell specialty.

      c.      Continuous monitoring and control are not required to maintain a culture of continuous improvement.

      d.      Resources are allocated equitably without distinction between processes and ranks of participants.

      e.      I don't know the answer.

B3.     Which of the following alternatives corresponds to the best definition of "5S"?

      a.      Technique focused on adapting work environments to make them organized and clean, thus improving productivity.

      b.      A method that allows finding the root cause of productivity problems.

      c.      Process of continuous improvement based on concrete, simple and inexpensive actions involving all the workers of an organization.

      d.      Five ways to select industrial processes.

      e.      I don't know the answer.

B4.     Which of the following alternatives corresponds to the correct order of application of "5S"?

      a.      SEISO (shine)—SEIRI (sort)—SEITON (set in order)—SEIKETSU (standardize)—SHITSUKE (sustain).

      b.      SEISO (shine)—SEITON (set in order)—SEIRI (sort)—SEIKETSU (standardize)—SHITSUKE (sustain).

      c.      SEIRI (sort)—SEITON (set in order)—SEISO (shine)—SEIKETSU (standardize)—SHITSUKE (sustain).

      d.      SEIRI (sort)—SEITON (set in order)—SEISO (shine)—SHITSUKE (sustain)—SEIKETSU (standardize).

      e.      I don't know the answer.

B5.     Which of the following alternatives corresponds to the definition of "cycle time"? (Please consider the airplane manufacturing process).

      a.      Time required to produce the total number of airplanes without defects.

      b.      Time required to produce one cycle (10 airplanes) without defects.

      c.      Time required to produce the first defect-free airplane.

      d.      None of the answers is correct.

      e.      I don't know the answer.

B6.     Which of the following alternatives corresponds to the definition of "WIP (Work in Progress)"? (Please consider the airplane manufacturing process).

      a.      A number of incomplete airplanes (progress was made, but not completed).

      b.      A number of airplanes were completed but with defects (these did not fly).

      c.      A number of unused sheets (requested but not used).

      d.      None of the answers is correct.

      e.      I don't know the answer.

B7.     Which of the following alternatives corresponds to the definition of "Inventory"? (Please consider the airplane manufacturing process).

      a.      A number of incomplete airplanes (progress was made, but not completed).

      b.      A number of airplanes were completed but with defects (these did not fly).

      c.      A number of unused sheets (requested but not used).

      d.     None of the answers is correct.

      e.     I don't know the answer.

C1.    Do you consider that this activity (game) reinforces the teaching-learning process? (Scale from 1: Strongly disagree to 5: Strongly agree).

      a.     Strongly disagree.

      b.     Disagree.

      c.     Neither agree nor disagree.

      d.     Agree.

      e.     Totally agree.

C2.    Would you like more activities (games) like this in your curriculum courses?

      a.     Yes.

      b.     No.

      c.     I don't know.

C3.    What did you like most about this activity (game)?

Answer: \_\_\_\_\_\_

C4.    What changes would you make (proposal for improvement)?

Answer: \_\_\_\_\_\_

**Appendix F. Informed Consent**

Dear Participant,

Your participation in the research project "Paper Airplanes in Engineering Education", is related to evaluating the understanding of production systems supported by Lean construction tools: Continuous Improvement Cells and 5S. For this reason, it is important to have your voice, opinion, and perception so that we can enrich the analysis of the information collected.

You are not obliged to allow the information generated related to you to be included in the analysis of the research. If you decide not to participate in the study, or if you decide to withdraw your information from the data set at any time, you will not be harmed. If you decide to collaborate with this project, your testimony will be included in the data to be analyzed in the research. The collection of information will be carried out in the development of the activities of the course, which may include:

– Allowing observation of course sessions.

– Participating in individual online surveys.

Your participation in this study is free and voluntary, and you may request to be excluded from this research and that your interventions not be considered in this research without prior justification or prejudice to you. You will be free not to respond to any of the questions and/or requests you receive in the framework of the generation of information, if it may cause you any harm or discomfort.

The data will be stored in the Google Drive platform for the duration of the research and the research team will have access to them. The data will be stored for two years, at the end of which time it will be deleted in a reserved manner.

Given the characteristics of the study, the data will be used only in academic research instances and for research dissemination. Likewise, the research team assumes a commitment to confidentiality to protect the identity of all those involved in this study.

All information you share in this study is completely confidential and your anonymity will be guaranteed. At no time will your name or identity be revealed, nor will any personally identifiable information about you or any of your students be revealed.

If you have any questions during or after the systematization of the course, you may contact the research team. By responding to this survey, you allow the use of your information in this study. Thank you for your attention.

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
