# Peer review of "Paper Planes for Teaching Construction Production Systems Based on Lean Tools: Continuous Improvement Cells and 5S"

_buildings, doi:10.3390/buildings13020558_

Round 1

Reviewer 1 Report

This paper is a well-written, well-structured, and comprehensive paper regarding "Teaching Construction Production Systems based on Lean Tools". The research method is well explained and supported by applying it in undergraduate and graduate courses. This manuscript has rich, beautiful diagrams, figures, and accurate data supporting conclusions. In addition, the structure of the manuscript is rigorous, and the language logic is reasonable. The subject is relevant to the building journal's audience; however, minor issues must be rectified.

The authors need to address the following comments:

1-  The search query for finding the papers in the literature must be included.

2-  Table 1 is very informative. However, It is preferable to include a column detailing which lean principles each game emphasizes.

3-     Some of the links in figure 3 should be corrected. For instance, The input of the definition solution objective should be derived from the analysis results of the preceding section. Also, some links need to be clarified.

4- This paper has been analyzed from several points, and its content is very rich; however, the final integration instructions still need to be included. There is little substance in the conclusion section, which is disconnected from the Discussion section. To enhance the cogency of the conclusions, it is possible to briefly integrate the findings of the discussion in the Conclusion section.

5- A few language problems on the paper need to be reviewed and modified.

Author Response

The authors wish to thank the Editors and Reviewers for their time and effort in reviewing our manuscript and for their suggestions and observations. We have addressed all their comments in the following itemized responses (item-by-item), providing detailed explanations, as well as the consequent modifications of the paper. The paper is much improved for the suggestions.

Reviewer 2 Report

Dear colleagues,

The title of the article is catchy and attracts the researchers’ attention.

1. On the whole, the article in general is more about the game teaching methodology than about construction, this is my main point

2. The number of lean construction tools is much more than the authors presented. The number of tools considered in the article is recommended to be expanded.

3. No semantic link (or low correlation) between paper airplanes and construction. The authors should make a clearer link between their research and construction sphere.

Author Response

The authors wish to thank the Editors and Reviewers for their time and effort in reviewing our manuscript, as well as for their suggestions and observations. We have addressed all their comments in the following itemized responses (item-by-item) providing detailed explanations, as well as the consequent modifications of the paper. The paper is improved for the suggestions.

Reviewer 3 Report

1- Abstract

The motive of the research is briefly highlighted. Method chosen and findings from the research are also included.

I would suggest including the theoretical and practical contribution in the abstract of the work.

2- Introduction

Line XX - Please cite this sentence – According to Liker and Meier,

the author [9], who is another author, was cited in the phrase.

9. Salazar, L.A.; Revuelta, M.P. Simulation Paper Planes a Way to Teach Lean Production. In Proceedings of the IEEE International Conference on Industrial Engineering and Engineering Management; IEEE Computer Society, December 14 2020; Vol. 2020-December, pp. 1078–1082.

Line XX - Please don't use too many references. games, and linguistic exercises, among others [8,9,17,19,20].

Line XX - Please avoid reference overkill - the courses' theoretical content [9,17,20,21].

Please revise the article entirely, replacing 5s to 5S.

It is recommended that the authors include the article overview in the last paragraph.

3- Literature background

Figure 1 – Any reference to the figure 1 or it was legitimately made by the authors?

Revised the following subheading to the subsection.

2. Literature background

2.1. Continuous Improvement Cells

2.1.2. 5s in construction

2.3. Games for education

Therefore, the subsection on the 5S items.

â–ª SEIRI (sort): The initial stage involves

2.2.1. SEIRI (sort):

2.2.2. SEITON (set in order):

4- Research Methods

Tables 2 and 4 are presented for what reason?

Table 2. Professors who participated in the problem identification workshop.

Table 4. Game evaluator professors.

Table 3. Could you provide references for the key performance indicators?

Author Response

(The authors gave the same response as above.)
